# Orbital climate variability on the northeastern Tibetan Plateau across the Eocene–Oligocene transition

Hong Ao [1,2,3,4✉], Guillaume Dupont-Nivet [5,6,7✉], Eelco J. Rohling [8,9], Peng Zhang [1,3], Jean-Baptiste Ladant [10], Andrew P. Roberts [8], Alexis Licht [11], Qingsong Liu[12], Zhonghui Liu [13], Mark J. Dekkers [14], Helen K. Coxall [15], Zhangdong Jin [1,2,16], Chunju Huang [4], Guoqiao Xiao [4], Christopher J. Poulsen [10], Natasha Barbolini [17], Niels Meijer [7], Qiang Sun [18], Xiaoke Qiang [1], Jiao Yao [1] & Zhisheng An [1,2,19✉]

The first major build-up of Antarctic glaciation occurred in two consecutive stages across the Eocene–Oligocene transition (EOT): the EOT-1 cooling event at ~34.1–33.9 Ma and the Oi-1 glaciation event at ~33.8–33.6 Ma. Detailed orbital-scale terrestrial environmental responses to these events remain poorly known. Here we present magnetic and geochemical climate records from the northeastern Tibetan Plateau margin that are dated precisely from ~35.5 to 31 Ma by combined magneto- and astro-chronology. These records suggest a hydroclimate transition at ~33.7 Ma from eccentricity dominated cycles to oscillations paced by a combination of eccentricity, obliquity, and precession, and confirm that major Asian aridification and cooling occurred at Oi-1. We conclude that this terrestrial orbital response transition coincided with a similar transition in the marine benthic $\delta^{18}O$ record for global ice volume and deep-sea temperature variations. The dramatic reorganization of the Asian climate system coincident with Oi-1 was, thus, a response to coeval atmospheric $CO_2$ decline and continental-scale Antarctic glaciation.

[1] State Key Laboratory of Loess and Quaternary Geology, Institute of Earth Environment, Chinese Academy of Sciences, Xi'an, China. [2] CAS Center for Excellence in Quaternary Science and Global Change, Chinese Academy of Sciences, Xi'an, China. [3] Open Studio for Oceanic-Continental Climate and Environment Changes, Pilot National Laboratory for Marine Science and Technology (Qingdao), Qingdao, China. [4] State Key Laboratory of Biogeology and Environmental Geology, School of Earth Sciences, China University of Geosciences, Wuhan, China. [5] Université de Rennes, CNRS, Géosciences Rennes, UMR, 6118 Rennes, France. [6] Key Laboratory of Orogenic Belts and Crustal Evolution, Peking University, Beijing, China. [7] Universität Potsdam, Institute of Geosciences, Potsdam, Germany. [8] Research School of Earth Sciences, Australian National University, Canberra, Australia. [9] Ocean and Earth Science, University of Southampton, National Oceanography Centre, Southampton, UK. [10] Department of Earth and Environmental Sciences, University of Michigan, Ann Arbor, MI, USA. [11] Department of Earth and Space Sciences, University of Washington, Seattle, USA. [12] Centre for Marine Magnetism (CM2), Department of Ocean Science and Engineering, Southern University of Science and Technology, Shenzhen, China. [13] Department of Earth Sciences, University of Hong Kong, Hong Kong, China. [14] Paleomagnetic Laboratory 'Fort Hoofddijk', Department of Earth Sciences, Faculty of Geosciences, Utrecht University, Utrecht, The Netherlands. [15] Department of Geological Sciences, Stockholm University, Stockholm, Sweden. [16] Institute of Global Environmental Change, Xi'an Jiaotong University, Xi'an, China. [17] Department of Ecosystem and Landscape Dynamics, Institute for Biodiversity and Ecosystem Dynamics, University of Amsterdam, Amsterdam, The Netherlands. [18] College of Geology and Environment, Xi'an University of Science and Technology, Xi'an, China. [19] Interdisciplinary Research Center of Earth Science Frontier, Beijing Normal University, Beijing, China. ✉email: aohong@ieecas.cn; guillaume.dupont-nivet@univ-rennes1.fr; anzs@loess.llqg.ac.cn

The Eocene–Oligocene transition (EOT) at ~34 Ma marks the main transition from the early Cenozoic greenhouse to the modern icehouse world[1]. Marine records and climate model experiments suggest that this key transition occurred in two stages[2–5]. The first stage (EOT-1) at ~34.1–33.9 Ma, in the upper portion of the reversed polarity chron C13r, is marked by moderate cooling and modest Antarctic ice volume increase[2–5]. The second, more dramatic, stage (Oi-1 glaciation) occurred around the C13r–C13n boundary at ~33.8–33.6 Ma and represents the culmination of the greenhouse-to-icehouse transition with ice-sheet expansion to the Antarctic coastline[2–5]. Oceanic and atmospheric circulation, ocean productivity, ocean carbonate compensation depth, and the global carbon cycle changed substantially from the late Eocene to early Oligocene[3–12]. Detailed reconstructions of climate variability across the EOT in the oceanic and continental realms, and in both the northern and southern hemispheres, help to elucidate the dynamics and interactions of large-scale climate changes in response to atmospheric $CO_2$ decrease and global cooling[3,6,9,13–15].

Terrestrial EOT records are relatively rare and include a few from East Asia (Maoming, Xining, Qaidam, and Junggar basins, and Mongolia)[9,14,16–23], central North America[13,24], southernmost South America[25], and Northern Europe[26] (Supplementary Fig. 1). These terrestrial records reveal long-term changes across the EOT, including a broad shift to cooler and drier continental climate that is generally consistent with ocean cooling[15,27–29]. However, they often lack sufficient temporal resolution to constrain associated short-term climate events and orbital-scale changes. As a result, the timing of the major terrestrial climate shift, and its relationship with the marine EOT-1 and Oi-1 events, remains uncertain. Continuous and expanded (>500 m thick) Eocene to Oligocene playa-palaeolake sequences on the NE Tibetan Plateau margin, including those in the Lanzhou and Xining basins (Fig. 1), allow high-resolution analysis of Tibetan EOT sequences. The EOT has been pinpointed stratigraphically in the Xining Basin, and has been associated with terrestrial cooling and aridification based on environmental magnetic, clay mineral, isotopic, palaeontological, and sedimentological changes[9,14,16–18]. These studies suggest that the terrestrial playa-palaeolake sequences on the NE Tibetan Plateau provide sensitive archives of past climate changes on land and have the potential to reveal details of orbital-scale terrestrial responses to global events through the EOT. At present, analysis of both marine[15,29] and terrestrial[9,13,14,16–21,23,25,26] records has focused primarily on general long-term cooling and/or aridification trends across the EOT, but much less on orbital-scale climate variability[4,9,11,12,16,30–32], because of a lack of sufficiently high-resolution palaeoclimate records that span continuously from the late Eocene to early Oligocene.

Here, we present environmental magnetic and elemental records for the ~35.5–31 Ma age interval from the Lanzhou Basin, located southeast of the Xining Basin on the NE Tibetan Plateau margin. They suggest an orbital variability shift for terrestrial Asian hydroclimate at ~33.7 Ma and constrain the major Asian aridification and cooling episode to coincide with the Oi-1 event. Based on terrestrial-to-marine correlations, we attribute these terrestrial climate changes to atmospheric $CO_2$ decline and major Antarctic glaciation at Oi-1.

## Results

### Setting, stratigraphy, and sedimentology of Eocene–Oligocene sediments in the Lanzhou Basin.
The NE Tibetan Plateau, both today and during the EOT, is situated in a transitional semi-arid region under the combined influence of the Asian summer and winter monsoons and the Westerlies, and is, therefore, sensitive to hydroclimate (moisture) changes[9,10,14] (Fig. 1). In the

Palaeogene, both the Lanzhou and Xining basins were part of the larger Longzhong Basin that resulted from slow subsidence after a Late Cretaceous fault initiation[33]. The Lanzhou Basin was later compartmentalized by fault reactivation related to the Indo-Asia collision[34,35] and is now bounded by the West Qinling mountains (dominated by Palaeozoic sedimentary rocks and Lower Palaeozoic plutonic rocks) in the south and the East Qilian Shan (dominated by Triassic submarine fan deposits and Permo-Triassic plutonic rocks) in the west and north[35] (Fig. 1; Supplementary Fig. 2). The Lanzhou Basin presents a relatively small area of roughly 300 km$^2$ that is filled with a thick (>1500 m) and mostly continuous sequence of Eocene–Miocene playa-palaeolake sediments that were transported primarily from the surrounding highlands[35]. Composed mostly of red fine-grained deposits, they provide an important early Eocene to Miocene regional climate archive[35–38].

The Duitinggou section (36°13′N, 103°37′E; 1,800 m elevation) studied here is located in the center of the Lanzhou Basin (Fig. 1). From older to younger, the Xiliugou, Yehucheng, and Xianshuihe Formations are recognized (Supplementary Fig. 3). The latest Eocene to earliest Oligocene succession studied here (383–570 m stratigraphic level within the Duitinggou section) is entirely within the Yehucheng Formation. The lithology comprises mudstone, siltstone, and fine sandstone successions with distinct gypsiferous cyclic intercalations (Supplementary Fig. 3). The overlying Oligocene–Miocene Xianshuihe Formation consists mainly of light red mudstones that are intercalated with sandstone or conglomerate packages[37,38], while the underlying early Eocene Xiliugou Formation consists of red massive sandstones (Supplementary Fig. 3).

Cyclic gypsiferous intercalations in the mudstone/siltstone are typical features of the Yehucheng Formation in the Lanzhou Basin (Fig. 2; Supplementary Fig. 3) and are similar to those in the coeval Mahalagou Formation in the nearby Xining Basin, which were modulated by orbital climate oscillations[9,16,32]. Gypsum beds are white or greyish-green, and vary in thickness between ca 0.5 and 4 m. There are two main gypsum facies: (1) gypsum beds that consist of millimetre- to centimetre-scale fine-grained gypsum laminae that are generally associated with variable amounts of reddish-brown/greyish-olive laminated mudstone/siltstone beds, and (2) gypsum beds that are dominated by decimetre- to metre-thick tabular or nodular beds of alabastrine gypsum with rarely preserved lacustrine laminations due to chickenwire structures, displacive enterolithic veins, and cracks. Cyclic gypsiferous intercalations indicate perennial subaqueous playa saline lake conditions with higher groundwater levels that allowed solutes to develop metre-scale gypsum layers[9,32]. In contrast, mudstone beds are reddish brown to dark red and massive. Notably, their red colour and widespread cm-scale slickensides (Supplementary Fig. 3) suggest that they have been subjected to occasional oxidizing conditions, i.e., a distal alluvial fan environment and ephemeral subaerial exposure with moderate pedogenesis under a low-gradient floodplain or dry distal mudflat environment[9]. The absence of colour mottling, gley-features, and carbonate nodules indicates that the regional palaeoenvironment was not sufficiently humid for intense pedogenesis and soil formation after deposition[32]. Greyish white siltstones are likely to have been deposited under subaqueous alluvial to shallow playa lake conditions, while light red siltstones possibly formed under alluvial to even shallower ephemeral playa lake conditions with occasional subaerial exposure. Field investigation suggests that both mudstone and siltstone beds are homogeneous throughout this section, without apparent changes from the lower to upper intervals, which is supported by the mean grain size record (Fig. 2). They occasionally have variable small amounts of dispersed fine-grained or lenticular gypsum grains, which may have originated from evaporating

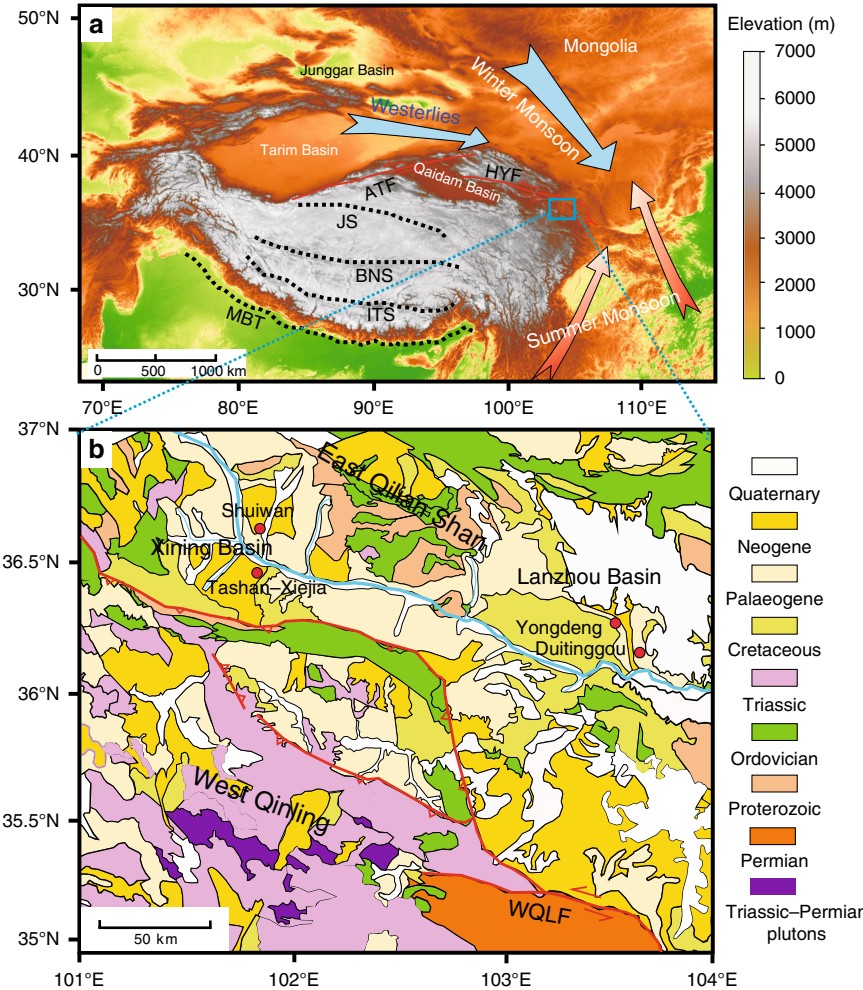

**Fig. 1 Site location. a** Topographic map of the present-day Tibetan Plateau region with relevant atmospheric features (indicated with blue and brown arrows) and major tectonic features (MBT, Main Boundary Thrust of the Himalaya; ITS, Indus-Tsangpo Suture; BNS, Bangong-Nujiang Suture; JS: Jinsha Suture; ATF, Altyn Tagh Fault; HYF, Haiyuan Fault). **b** Simplified geological map of the Lanzhou and Xining basins with the West Qinling Fault (WQLF) and locations of sections (red solid circles) mentioned in the text.

phreatic groundwater, occasional surface waters percolating into the sediment, and/or riverine input of previously formed gypsum grains[32]. In addition, four 1–3 m thick greyish-white fine sand beds were deposited in the Yehucheng Formation under subaqueous alluvial to shallow playa lake conditions. They have larger mean grain size values than the (gypsiferous) siltstone and mudstone beds (Fig. 2). Potential bioturbational or pedogenic mixing was not significant as suggested by massive mudstones, often laminated gypsum layers, siltstones with clear linear bedding, rare centimetre-scale crossbedding, and absence of root and burrow traces. Similar to comparable deposits in the Xining Basin, we infer that sediment deposition occurred in distal alluvial playa-palaeolake environments: gypsum intervals formed during perennial subaqueous playa saline lake conditions, greyish white or light red siltstone intervals formed during shallow playa lake to alluvial anoxic subaqueous conditions, and red mudstone intervals formed during distal alluvial fan conditions[9,16,32,39]. The gypsum and siltstone intervals corresponding to perennial anoxic alluvial to playa lake conditions may relate to higher water supply than the mudstone intervals that correspond to oxic distal alluvial fan conditions[9,16,32,39]. At the 502-m level in the Duitinggou section, a notable absence of gypsum beds indicates a marked environment shift (Fig. 2). In the neighbouring Xining Basin, a similar change occurred during the EOT, and is

interpreted to indicate a groundwater level decrease associated with regional aridification[9,16,32].

**Late Eocene to early Oligocene magneto-astrochronology from the Lanzhou Basin.** Constrained by mammal and pollen biostratigraphy, the Xianshuihe Formation magnetostratigraphy[37,38] in the Duitinggou section (0–383 m) has been documented to range from polarity chron C5En down to uppermost chron C12r, with ages from ~18 to 31 Ma (Supplementary Fig. 4). This chronology is supported by consistent correlations with biostratigraphic and magnetostratigraphic records from the neighbouring Xining Basin[34,40] (Supplementary Fig. 4). Combining the established chronology of the concordantly overlying Xianshuihe Formation[37,38], detailed palaeomagnetic analysis (Supplementary Note 1; Supplementary Figs. 4–6) provides a downward magnetostratigraphic continuation from polarity chrons C12r to C16n.1n for the Yehucheng Formation, with magnetostratigraphic ages of ~31–35.7 Ma according to the 2012 geomagnetic polarity time scale (GPTS)[41] (Fig. 2). Consistent with the Yongdeng section (~40 km northwest of the Duitinggou section) in the northern Lanzhou Basin[36], our data suggest that the top of reversed polarity chron C12r falls within a basin-wide yellow sand layer (marker layer A), which contains an early Oligocene mammal fauna and marks the boundary between the Yehucheng and Xianshuihe Formations

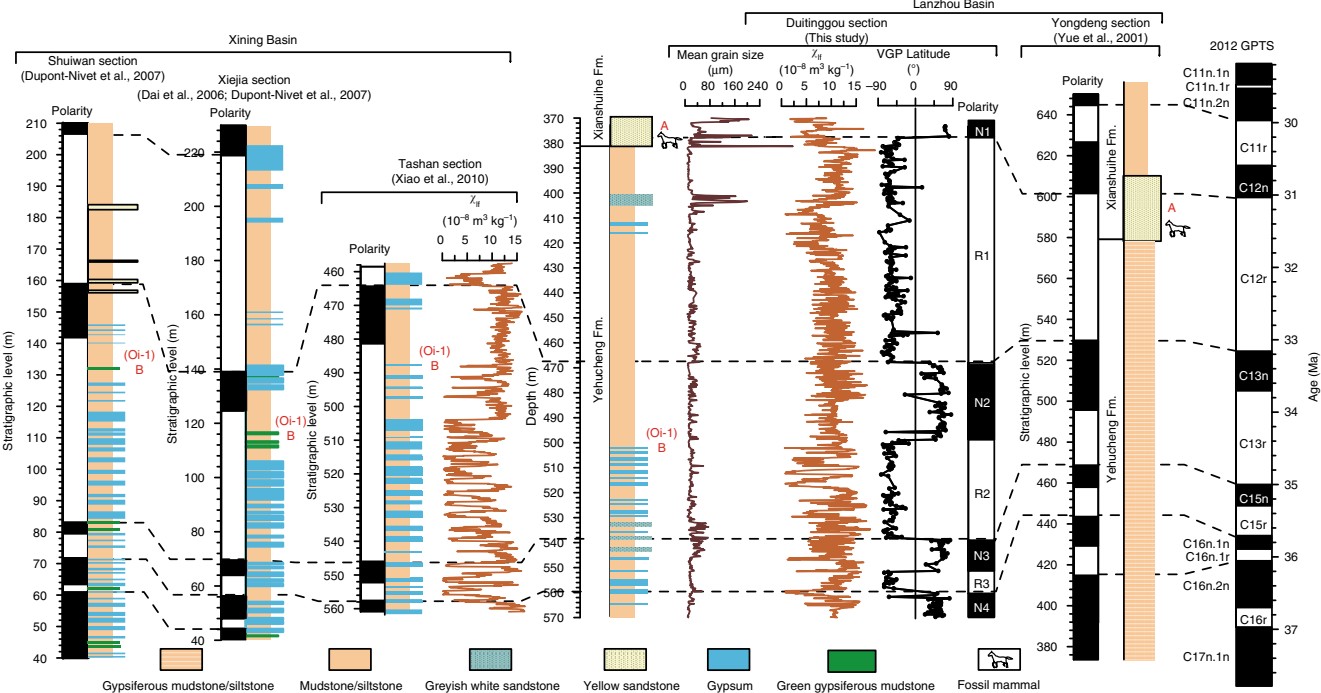

**Fig. 2 Regional lithostratigraphic and magnetostratigraphic correlations.** Lithostratigraphy and magnetostratigraphy of the Duitinggou and Yongdeng[36] sections in the Lanzhou Basin are compared with those of the Shuiwan[9], Tashan[16], and Xiejia[9,40] sections in the Xining Basin, and correlate with the 2012 geomagnetic polarity time scale (GPTS)[41]. Mean grain size and magnetic susceptibility ($\chi_{lf}$) records for the Duitinggou section and the Tashan $\chi_{lf}$ record[16] are also shown. The uppermost 14 virtual geomagnetic pole (VGP) latitudes (open circles) for the Duitinggou section are from the bottom of the Xianshuihe Formation[37]. The yellow sand layer (layer A) that overlies the Yehucheng Formation at the Duitinggou and Yongdeng sections is a prominent marker layer across the Lanzhou Basin. The Oi-1 event coincides with the end of regular alternations of gypsum and red mudstone/siltstone beds at gypsum (or green gypsiferous mudstone) layer B, and with a decreased amplitude variability of $\chi_{lf}$ at 100-kyr eccentricity periods. The Yehucheng Formation in the Yongdeng section was described only as gypsiferous mudstone/siltstone; more detailed lithological variations were not reported by Yue et al.[36].

(Fig. 2). This supports the correlation of the thick reversed polarity zone R1 of the upper Yehucheng Formation underlying marker layer A to reversed polarity chron C12r. In both the Lanzhou and Xining basins, regularly cyclic gypsum beds were present during the late Eocene and disappeared just prior to normal polarity chron C13n. This supports the correlation of thick reversed polarity zone R2 to reversed polarity chron C13r. Regional stratigraphic correlations, biostratigraphic constraints, and clear correlation with the GPTS together indicate that the normal polarity zone N2 between R1 and R2 correlates with normal polarity chron C13n (Fig. 2; Supplementary Note 1; Supplementary Fig. 4).

**Palaeoclimatic records of the EOT from the Lanzhou Basin.** Detailed magnetic and mineralogical analyses (Supplementary Figs. 5 and 7–12) suggest that low-frequency magnetic susceptibility ($\chi_{lf}$) of playa-palaeolake deposits in the Lanzhou Basin reflects primarily magnetic mineral concentration changes (see Supplementary Note 2 for details). The $\chi_{lf}$ record has a higher resolution than the saturation isothermal remanent magnetization (SIRM), and hard isothermal remanent magnetization (HIRM) records, but they all vary consistently throughout the section (Supplementary Fig. 12). As is the case in the Xining Basin[9,16], their values are low in gypsum and siltstone layers, which represent perennial playa lake to alluvial anoxic subaqueous conditions with high precipitation. Such conditions probably facilitated partial post-depositional magnetite and hematite dissolution and gypsum formation that diluted the magnetic expression. They also potentially drove rapid clastic material transportation to the playa lake from catchment regions; potential subaerial exposure had shorter durations that limited pedogenic processes during and after deposition. This would have limited pedogenic magnetic mineral

formation. In contrast, the $\chi_{lf}$ values are high in red mudstone beds, which represent lake retreat and oxic distal alluvial fan subaerial conditions with low precipitation. Such conditions would have increased pedogenic magnetic mineral formation during increased ephemeral subaerial exposure. The presence of pedogenic superparamagnetic (SP) particles is suggested by first-order reversal curve (FORC) diagrams[42] and a positive linear correlation between frequency-dependent magnetic susceptibility ($\chi_{fd}$) and $\chi_{lf}$ (Supplementary Figs. 9, 10). Oxic distal alluvial fan conditions also facilitated detrital magnetic mineral preservation during deposition. Moreover, gypsum beds disappeared and dilution of non-magnetic materials decreased substantially (see Supplementary Note 2 for details; Supplementary Fig. 13).

Thus, combined pedogenic, dissolution, dilution, and preservation effects provide a plausible model to explain notable orbital-scale $\chi_{lf}$ changes of the Lanzhou Basin fluvial-lacustrine sediments. As observed in sedimentary, clay mineral, and pollen studies in the Xining Basin[9,14,16–18], low $\chi_{lf}$ units in the Lanzhou Basin, which are indicative of low magnetic mineral concentrations, are also related to high regional precipitation based on our likely combined mechanism, and vice versa (Supplementary Fig. 13). With the $\chi_{lf}$ record providing a past hydroclimate proxy, we further refined the magnetochronology by tuning the 405-kyr and 100-kyr components in our $\chi_{lf}$ record to Earth's computed orbital eccentricity record[43] (see "Methods" for details; Supplementary Figs. 14–16). Below we assess NE Tibetan Plateau terrestrial climate evolution across the EOT using environmental magnetic and element-based chemical weathering records using our refined astronomical time scale.

The $\chi_{lf}$, SIRM, and HIRM records are all characterized by a shift at ~33.7 Ma (Fig. 3c–e), which coincides with a lithological

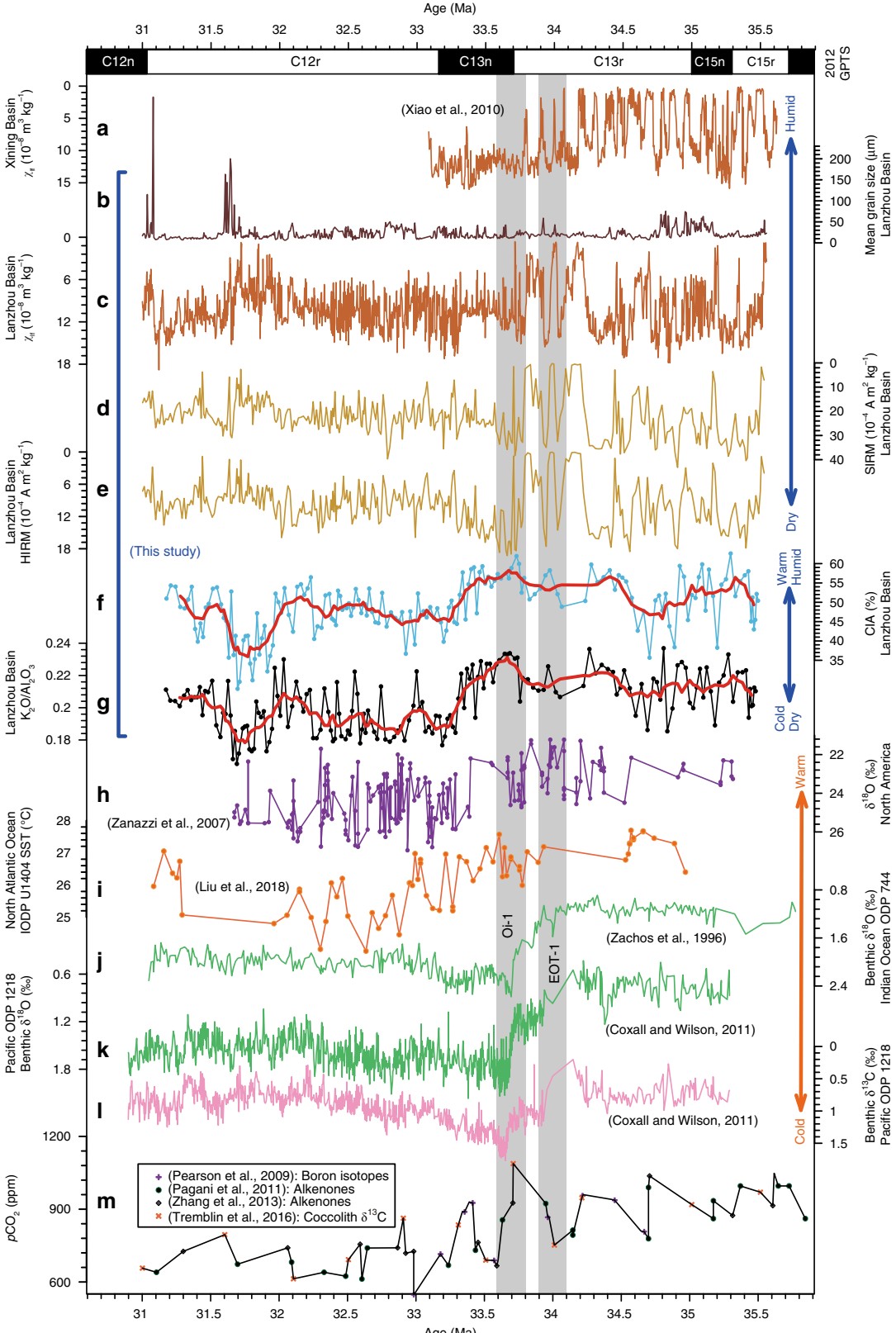

shift (Fig. 2). Consistently higher amplitude variability is observed with a 100-kyr eccentricity period before ~33.7 Ma (Fig. 3c–e), which is also notable in the Xining Basin $\chi_{lf}$ record (Fig. 3a). The long-term $\chi_{lf}$ time series of the Duitinggou section has an average resolution of 2.5 kyr between 31.0 and 35.5 Ma, with a typical resolution between ~2 and 5 kyr before 33.7 Ma and between ~1 and 4 kyr after 33.7 Ma (Supplementary Fig. 16h). The resolution is sufficient to assess orbital variability in the obliquity band (41-kyr), while more tentative indications can be obtained for the precession band (~23-kyr). Spectral analysis of the $\chi_{lf}$ record

**Fig. 3 Global and terrestrial climate changes across the Eocene–Oligocene transition (EOT). a** Low-frequency magnetic susceptibility ($\chi_{lf}$) for the Tashan section, Xining Basin[16]. **b** Mean grain size, **c** $\chi_{lf}$, **d** saturation isothermal remanent magnetization (SIRM), **e** hard isothermal remanent magnetization (HIRM), **f** chemical index of alteration (CIA), and **g** $K_2O/Al_2O_3$ records from Lanzhou Basin playa-palaeolake sediments. Age models for both the Lanzhou and Xining records were established first by magnetochronology and were then refined by matching 405-kyr and 100-kyr components in the $\chi_{lf}$ record with the computed record of Earth's orbital eccentricity[43]. **h** $\delta^{18}O$ record of fossil remains from central North America[13]. **i** Sea surface temperature (SST) from IODP U1404, North Atlantic Ocean, based on alkenone unsaturation index[15]. **j** Benthic foraminiferal $\delta^{18}O$ from ODP Site 744, southern Indian Ocean[52]. **k, l** Benthic foraminiferal $\delta^{18}O$ and $\delta^{13}C$ from ODP Site 1218, tropical Pacific Ocean[5]. The two increasing $\delta^{18}O$ and $\delta^{13}C$ steps are designated as EOT-1 and Oi-1, respectively. **m** Atmospheric $pCO_2$ reconstruction across the EOT from ODP Site 925/929 with coccolith $\delta^{13}C$, boron isotopes, and alkenones[53–56].

suggests strong expression in the 405-kyr and 100-kyr bands throughout the ~35.3–31.3 Ma interval (Fig. 4a–c). Notably, after ~33.7 Ma, there is a strong obliquity expression, with a relatively weaker precession expression, and the obliquity amplitude-modulated 1.2 Myr band is observed superimposed on the 100 and 405-kyr bands (Fig. 4a–c). Similar orbital variability is observed in the spectral evolution of 3-point and 5-point running $\chi_{lf}$ means, which suppresses potential short-term noisy signals (Supplementary Fig. 17). These orbital climate variations, particularly the orbital expression shift at 33.7 Ma, are also evident in the depth domain and in the untuned magnetochronology (Supplementary Figs. 14, 15). This confirms the robustness of the orbital response shift of the Lanzhou Basin climate identified across Oi-1. We note that in the untuned magnetochronology the calculated eccentricity, obliquity, and/or precession bands are displaced slightly or have a subdued expression in a few intervals where a non-orbital signal structure appears to be more prominent (Supplementary Fig. 15). In our refined astronomical time scale, orbital expression (eccentricity, obliquity, and precession) in the $\chi_{lf}$ record is enhanced significantly, and non-orbital noise is substantially lower (Fig. 4c) relative to the untuned magnetochronology (Supplementary Fig. 15).

The chemical index of alteration (CIA, the molar ratio of $Al_2O_3$ to $Al_2O_3 + CaO + Na_2O + K_2O$) and the $K_2O/Al_2O_3$ ratio are two classic chemical weathering proxies; they correlate positively with chemical weathering intensity[44,45]. They vary consistently during the late Eocene–early Oligocene (Fig. 3f–g), and correlate linearly with each other (Supplementary Fig. 18). While the CIA and $K_2O/Al_2O_3$ records have much lower sampling resolution and cannot capture detailed orbital variability like the high-resolution $\chi_{lf}$ record, some 100-kyr cycles are roughly evident in both chemical weathering records before ~34.6 Ma and after ~32.8 Ma (Fig. 3f–g). Notably, the negative CIA and $K_2O/Al_2O_3$ shifts from ~33.7 to 33.2 Ma indicate a substantial chemical weathering intensity decrease in the Lanzhou Basin (Fig. 3f–g), consistent with a change from more intensive weathering under higher temperature and precipitation conditions to insignificant alteration under lower temperature and precipitation conditions across the Oi-1 event[44,45]. The observed shifts cannot be explained in terms of grain size changes because CIA and $K_2O/Al_2O_3$ correlate poorly with mean grain size (Supplementary Fig. 18d, e). Moreover, grain size does not shift at 33.7–33.2 Ma (Fig. 3b).

## Discussion

Our high-resolution records indicate that orbitally paced variability was evident for the late Eocene–early Oligocene NE Tibetan Plateau climate. Consistent with the Lanzhou Basin $\chi_{lf}$ record (Fig. 4a–c), spectral analysis of the high-resolution Pacific benthic $\delta^{18}O$ record also indicates a strong eccentricity (405- and 100-kyr) band throughout the late Eocene–early Oligocene between ~35.1 and 31.3 Ma, and a superimposed strong obliquity and obliquity amplitude-modulated band of 1.2 Myr after 33.7 Ma, together with a relatively weaker and discontinuous precession expression (Fig. 4d–f). In the benthic marine $\delta^{18}O$ record, strong 405 and 100-kyr eccentricity bands persist continuously through at least the

Middle Miocene[11,30,31]. The Xining Basin $\chi_{lf}$ record has 22 regular large-amplitude cycles from polarity chrons C15r to C13r (Fig. 3a), with durations of 1.666 Myr (35.404–33.738 Ma) in the 2004 GPTS[46] and 2.001 Myr (35.706–33.705 Ma) in the updated 2012 GPTS[41], respectively. The 22 $\chi_{lf}$ cycles correspond to a 76-kyr periodicity based on the 2004 GPTS[46], which were interpreted previously as due to eccentricity[9], obliquity[16], or a combination of obliquity and eccentricity cycles[32]. However, these cycles have a 91-kyr periodicity in an age model based on the 2012 GPTS[41], which is consistent with eccentricity cycles. Spectral analyses of the Xining Basin $\chi_{lf}$ record using the refined astronomical chronology suggest pronounced eccentricity (100 and 405-kyr) rhythms between 35.3 and 33.3 Ma (Supplementary Fig. 19). Furthermore, the Maoming Basin (South China) lithological record has strong latest Eocene eccentricity cycles[19]. Thus, the Lanzhou Basin $\chi_{lf}$ and Pacific benthic $\delta^{18}O$ records suggest that NE Tibetan Plateau and global changes were consistently paced by eccentricity both before and after the onset of major Antarctic glaciation, consistent with the heartbeat of the Earth system as paced by eccentricity cycles[11,30,31,47].

Appearance in spectral analyses of strong obliquity cyclicity and of a 1.2-Myr obliquity-amplitude-modulated band after 33.7 Ma in both the Lanzhou Basin $\chi_{lf}$ and Pacific benthic $\delta^{18}O$ records (Fig. 4) indicates that Asian climate and Antarctic ice sheets started to respond dynamically to obliquity forcing across Oi-1. Likewise, strong obliquity cyclicity was also absent before 33.7 Ma and appeared clearly in the Xining Basin $\chi_{lf}$ record after this time (Supplementary Fig. 19). Cyclostratigraphy in the Maoming Basin lithology[19] seems to suggest a similar shift from dominantly eccentricity to obliquity cycles across Oi-1. Accordingly, it appears that this obliquity response was not significant in the latest Eocene when major Antarctic ice sheets were not established. In addition to obliquity cyclicity, the precession signal also appeared clearly after ~33.7 Ma in the spectral analyses of the Lanzhou Basin $\chi_{lf}$ record (Fig. 4). This orbital shift also seems to be present in the Pacific benthic $\delta^{18}O$ record[5], although the precession band is weaker and less continuous (Fig. 4). The resolution of the Lanzhou Basin $\chi_{lf}$ record (Supplementary Fig. 16h) is in principle sufficient to reveal the precession band before 33.7 Ma. While detailed sedimentary observations did not yield evidence for hiatuses in the late Eocene–early Oligocene Duitinggou section, occasional unidentified small hiatuses or sedimentation rate decreases may cause one or several precession cycles to be (partially) missed in a few intervals. However, they are unlikely to have been missed throughout the whole ~2-Myr long interval between 35.5 and 33.7 Ma (Supplementary Note 3), especially when a clear precession expression is observed after 33.7 Ma. Thus, such a precessional expression shift across the Oi-1 event in the Lanzhou Basin $\chi_{lf}$ record is unlikely related to sedimentation rate changes. We note, however, that validation from even more highly resolved multiple proxy records would be ideal.

Environmental smoothing related to post-depositional diagenesis and biological disturbance[48–50] is another potential factor that can influence orbital variability expressions in low sedimentation

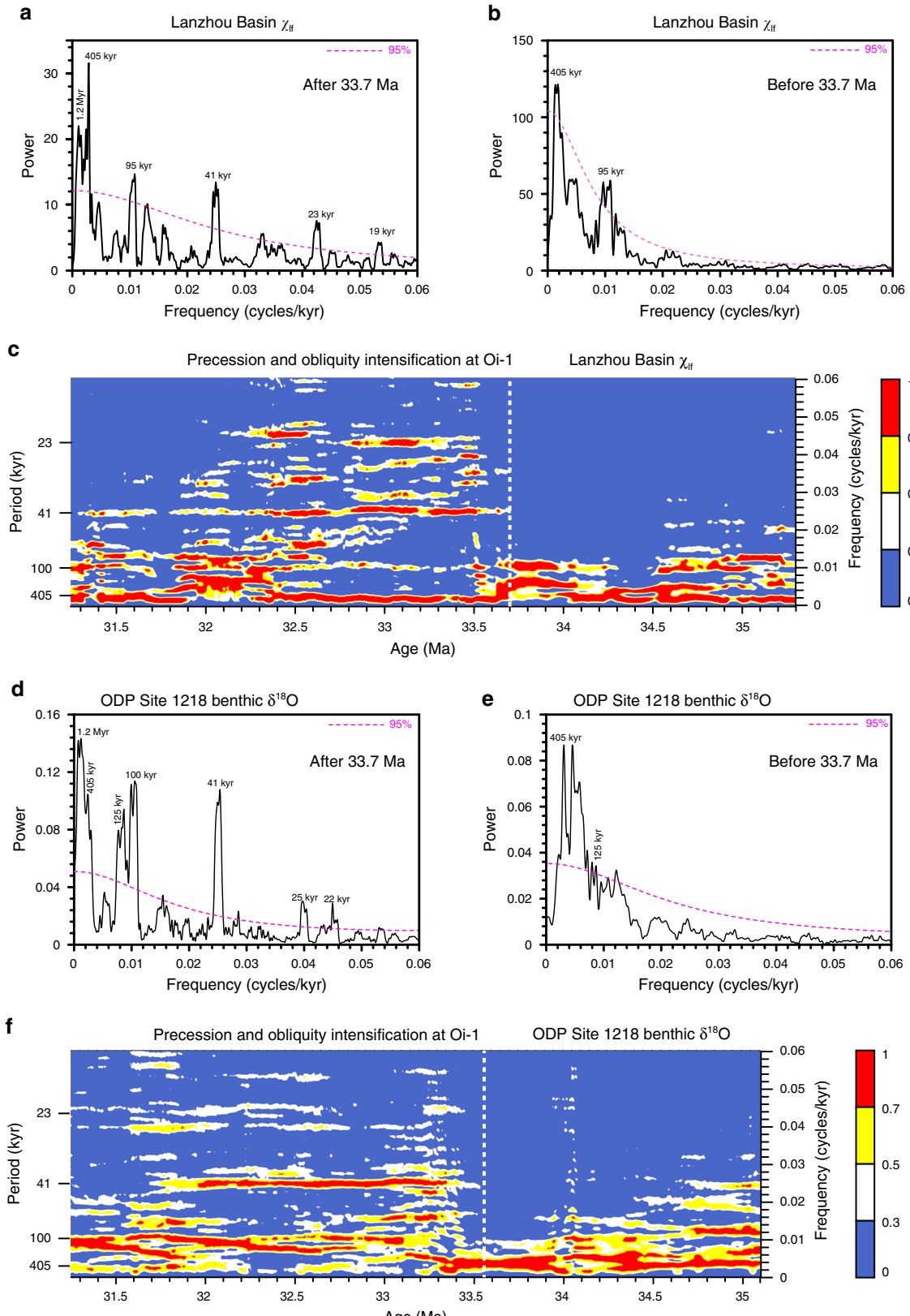

**Fig. 4 Orbital climate variability across the Eocene–Oligocene transition.** $2\pi$-Multi-taper method (MTM) power spectrum of (**a**, **b**) Lanzhou Basin $\chi_{lf}$ and (**d**, **e**) ODP Site 1218 benthic $\delta^{18}O$ records after and before 33.7 Ma, with a robust red-noise model at the 95% confidence level. Spectral evolution for (**c**) Lanzhou Basin $\chi_{lf}$ and (**f**) ODP Site 1218 benthic $\delta^{18}O$ records calculated using the MATLAB Evoffte routine, with a 600-kyr sliding window and 2-kyr sliding step.

rate records. However, it is unlikely to have affected significantly the Lanzhou playa-palaeolake sediments as suggested by the absence of root and burrow marks and by the preservation of centrimetric laminar mudstone and gypsum beds before 33.7 Ma (Supplementary Fig. 3d–g). Moreover, bioturbation is largely restricted to a narrow depth of surficial sediments, which varies from <3 cm for varved and laminated lake sediments to ~12 cm for homogenous massive lake sediments in wet regions[49,50]. Thus, bioturbational mixing depths for Lanzhou palaeolake sediments in arid western China with low organic content are expected to be smaller, probably one to two orders of magnitude smaller than the expected thickness of precession (70–100 cm) and obliquity (150–190 cm) cycles (Supplementary Fig. 14). In addition, evident precession and obliquity cycles are absent between 35 and 33.7 Ma from terrestrial records from Xining (Supplementary Fig. 19) and Maoming[19] basins, the marine ODP Site 1218 benthic $\delta^{18}O$ record (Fig. 4), and the high-resolution X-ray fluorescence (XRF) core scanning Si, Ca, and Fe records[47] for ODP Sites 1218, U1333, and U1334. All of these terrestrial and marine records are instead dominated consistently by eccentricity. More detailed discussion in Supplementary Note 3 documents that the absence of apparent precession and obliquity expressions between 35 and 33.7 Ma in the Lanzhou Basin $\chi_{lf}$ record is unlikely to have been caused by sedimentation rate/environmental smoothing variations or by unidentified small hiatuses. As summarized in a flowchart of our orbital tuning strategy (Supplementary Fig. 20), we conclude that the orbital response shift identified at ~33.7 Ma in the Lanzhou Basin $\chi_{lf}$ record is a robust feature of Asian climate reorganization across Oi-1, which is supported by (1) the Maoming cyclostratigraphy[19] and spectral analyses of (2) the Lanzhou Basin $\chi_{lf}$ record in the depth domain (Supplementary Fig. 14), and in astronomical and magnetostratigraphic time scales (Fig. 4; Supplementary Fig. 15), (3) the Xining $\chi_{lf}$ record (Supplementary Fig. 19), and (4) the ODP Site 1218 benthic $\delta^{18}O$ record (Fig. 4). In addition, a similar hydroclimate transition from eccentricity to combined eccentricity, obliquity, and precession pacing is observed in the Qaidam Basin, NE Tibetan Plateau, at ~8.5 Ma when the global climate cooled and Antarctic glaciation intensified[51].

The long-term trend in our records confirms previous inferences about Asian aridification and cooling across the EOT[9,14,16–21,23]. The higher resolution of our records constrains more precisely the age of the main transition to ~33.7 Ma, around the C13r–C13n boundary, which corresponds with the Oi-1 event (33.8–33.6 Ma) in high-resolution marine benthic $\delta^{18}O$ and $\delta^{13}C$ records[5,52], and post-dates the initial EOT-1 event at 34.1–33.9 Ma (Fig. 3). Sharp termination of periodic gypsum deposition at around the C13r–C13n boundary at ~33.7 Ma in the Lanzhou Basin and ~33.8 Ma in the Xining Basin, along with a shift to lower $\chi_{lf}$, SIRM, and HIRM minima, suggest a major shift to lower precipitation and to a smaller and/or shallower lake exactly during Oi-1 (Fig. 3). Earlier and less substantial aridification is suggested by decreasing gypsum bed thicknesses and associated low-$\chi_{lf}$ peaks at ~34.07 Ma in the Xining Basin possibly in correspondence with the EOT-1 event[16], which is not clearly expressed in the Lanzhou Basin records. This early change likely was more subtle than the later, region-wide change at 33.8–33.7 Ma, so the stronger Xining Basin expression compared to the Lanzhou Basin may reflect local variations in climate drivers and/or basin sensitivity to these drivers. In contrast, the later change at Oi-1 was large enough to produce unambiguous, major regional impacts. Consistent with the timing of major lithological and environmental magnetic shifts in the Lanzhou Basin, the negative CIA and $K_2O/Al_2O_3$ shifts that started at ~33.7 Ma and that were amplified at ~33.3 Ma indicate a prominently decreasing chemical weathering trend with drier and cooler terrestrial climate starting at the Oi-1 event (Fig. 3f–g), which was possibly accompanied by aeolian input increases[32]. The

CIA and $K_2O/Al_2O_3$ records have much lower sampling resolution and, thus, cannot be compared directly with the $\chi_{lf}$ record over short orbital (precessional) time scales for some intervals. However, these low-resolution chemical weathering proxies provide interpretable long-term trend changes consistent with that of the $\chi_{lf}$ record, which is supported by positive correlations of $\chi_{lf}$ with CIA and $K_2O/Al_2O_3$ (Supplementary Fig. 18b, c).

Pronounced NE Tibetan Plateau hydroclimate changes during Oi-1, including an orbital response shift and enhanced aridification and cooling, may have been linked to the coeval global climate reorganization. Although $CO_2$ reconstructions[53–56] lack the resolution and precise chronological constraints needed for detailed comparison with our records, they suggest that $CO_2$ probably dropped by >300 ppm during Oi-1, which is more than during EOT-1 (Fig. 3m). Ice-sheet-climate modelling results[2,57,58] indicate that the major $CO_2$ drop during Oi-1 may have been a primary driver for Antarctic ice-sheet expansion to the coastline at ca 33.7 Ma[3].

We link the marked climatic transition on the NE Tibetan Plateau that started during Oi-1 to the associated larger $CO_2$ drop and full Antarctic glaciation. Accordingly, atmospheric $CO_2$-driven global cooling[54,57] resulted in less atmospheric moisture under lower temperatures and weaker summer monsoonal circulation[10], which would have decreased moisture transport to the NE Tibetan Plateau by the summer monsoon from the northern Indian and western Pacific Oceans, and by the Westerlies from the proto-Parathethys Sea that extended over Eurasia during this time[32]. Palaeoclimate reconstructions and modelling results both suggest that Antarctic ice-sheet growth would have caused southern hemisphere Westerlies to intensify and shift northward to increase cold Southern Ocean bottom and intermediate water transportation to northern hemisphere ocean basins[3,6,7,59]. The large (>50 m) global mean sea-level lowering[3] due to major Antarctic glaciation would have also exposed northern Indian and western Pacific Ocean continental shelves, and caused westward retreat of the giant, shallow proto-Parathethys Sea. This large-scale oceanic retreat decreased moisture supply to the NE Tibetan Plateau[9,60] and led to more continental central Asian climates with cooler winters[61,62]. The contemporaneity of major Antarctic glaciation and sea-level lowering during Oi-1 with rapid Asian aridification is supported by the sharp termination of periodic gypsum beds in both the Lanzhou and Xining basins. The subsequent long-term trend of Antarctic deglaciation and sea-level rise from ~33.6 to 31.7 Ma may have driven partial recovery of NE Tibetan Plateau precipitation as indicated by a modest long-term decline in the Lanzhou Basin $\chi_{lf}$, SIRM, and HIRM records (Fig. 3c–e).

After their development at Oi-1, extensive Antarctic ice sheets during the lower-$CO_2$ early Oligocene responded dynamically to high-latitude southern hemisphere summer insolation variations with strong precession and obliquity cycles. Under the competing influence of the Asian summer monsoon and Westerlies[9,10], moisture availability in the semi-arid Lanzhou Basin varied consistently with orbital Antarctic ice-sheet forcing. Although the environmental background differed significantly from Quaternary glacial-interglacial cycles, which were dominated by both Antarctic and northern hemisphere ice-sheet variability, Oligocene glacial-interglacial Antarctic ice-sheet advances and retreats over precession and obliquity cycles in response to summer insolation may have influenced orbital variability of Asian terrestrial climate by potential variations in sea level, atmospheric and oceanic circulation, $CO_2$, and temperature[3,11]. Accordingly, during glacial intervals, lowered sea level would have increased the land area, thus lengthening the moisture transportation pathway of the Asian summer monsoon from the northern Indian and western Pacific Oceans to the NE Tibetan Plateau. In particular,

the proto-Paratethys Sea retreat may have lengthened markedly the transport pathway of the Westerlies[9,60]. Both would have reduced moisture transport to the NE Tibetan Plateau. In addition, lower $CO_2$ levels and temperatures would have reduced atmospheric water vapour formation in the northern Indian and western Pacific Oceans and the proto-Paratethys Sea during glacials. Accordingly, a decreased oceanic moisture supply during Oligocene glacials also played a role. These large-scale processes may have reversed during interglacials and led to relatively wetter climates. Thus, we infer that links of Asian climate variations with global climate system were enhanced after the Oi-1 event not only because of sensitivity to larger orbitally driven Antarctic ice-sheet variations, but also because of increased sensitivity to regional insolation forcing in a lower-$CO_2$ Oligocene icehouse world.

In summary, we conclude that, coinciding closely with the Oi-1 event, precipitation variability on the NE Tibetan Plateau shifted from being dominantly eccentricity-paced to responding to combined eccentricity, obliquity, and precession forcing. Such an orbital response shift is also evident for the wider global climate as suggested by the marine benthic $\delta^{18}O$ record. The NE Tibetan Plateau also experienced a major increase in aridification and cooling during Oi-1. We relate these pronounced changes in orbital climate variability, aridification, and cooling on the NE Tibetan Plateau to changed climatic boundary conditions across Oi-1, including a major $CO_2$ drop and development of continental-scale Antarctic ice sheets, which reorganized atmospheric and oceanic circulation along with global climate, including the terrestrial Asian climate system.

## Methods

**Sampling.** To obtain samples that were as fresh as possible and to decrease the potential influence of recent weathering, the weathered outcrop surface was removed (at least the topmost 20 cm) before collecting samples from freshly exposed sediment. For magnetostratigraphic analysis, 510 block samples were collected at 30–40-cm stratigraphic intervals and were oriented in the field with a compass. Two cubic samples (2 cm × 2 cm × 2 cm) were taken from each oriented block for thermal demagnetization treatment to establish a magnetochronology. Some leftovers of these block samples were further used for mineral magnetic and elemental measurements. A total of 2001 unoriented samples were collected for climate proxy measurements at 10-cm intervals. All experiments were carried out at the Institute of Earth Environment, Chinese Academy of Sciences, Xi'an, China.

**Palaeomagnetic analyses.** Stepwise thermal demagnetization of the natural remanent magnetization (NRM) was conducted using a TD-48 thermal demagnetizer. 510 oriented cubic samples (one per level) were stepwise heated at 18 successive steps with 10–50 °C temperature increments to a maximum temperature of 680 °C. After each demagnetization step, the remaining NRM was measured with a 2-G Enterprises Model 755-R cryogenic magnetometer housed in a magnetically shielded space. The NRM intensity of samples is usually of the order of $10^{-3}$–$10^{-2}$ A/m; the instrument background (or noise) level in the magnetometer is $<10^{-6}$ A/m. Demagnetization results were evaluated using orthogonal diagrams[63]; the principal component direction for each sample was computed using least-squares linear fitting[64]. Principal component analysis (PCA) was performed using the PaleoMag software[65]; PCA fits were not anchored to the origin of orthogonal diagrams[66].

Detailed palaeomagnetic analysis of stepwise thermal demagnetization results enabled us to construct a robust magnetostratigraphic chronology for the Duitinggou section. After removal of a secondary overprint isolated by progressive demagnetization to 250–350 °C (sometimes up to 400–450 °C), a characteristic remanent magnetization (ChRM) was isolated up to 680 °C (Supplementary Fig. 5). ChRM directions were determined using strict selection criteria. At least four (but typically 8–15) consecutive demagnetization steps that decay linearly toward the origin of orthogonal diagrams were used to determine the ChRM direction from 250–350 °C to 680 °C (sometimes from 400–450 °C to 680 °C), with maximum angular deviation (MAD) values ≤15° for line fits (not anchored to the origin). Polarity zones were defined here using at least three successive virtual geomagnetic pole (VGP) latitudes of identical polarity, which were calculated from ChRM directions. A few intervals with only a single palaeomagnetic direction may indicate a short-lived geomagnetic anomaly with ambiguous origin (cryptochrons) or a geomagnetic excursion[60,67], which need to be documented as a global feature before being considered as real, and were thus not used for determining the polarity zone and for calculating the overall mean direction. A few samples possibly

recorded a transitional geomagnetic field, with large ChRM direction divergences from the mean, and were also not used for polarity zone determination.

**Mineral magnetic measurements.** All 2001 unoriented samples were powdered and were then packed into non-magnetic cubic boxes (2 cm × 2 cm × 2 cm) for low-frequency magnetic susceptibility ($\chi_{lf}$) and high-frequency magnetic susceptibility ($\chi_{hf}$) measurements in the laboratory with a Bartington Instruments MS2 magnetic susceptibility meter. $\chi_{lf}$ and $\chi_{hf}$ were measured at 470 and 4700 Hz, respectively. Frequency-dependent magnetic susceptibility ($\chi_{fd} = \chi_{lf} - \chi_{hf}$) was calculated. We selected 363 samples at a 50 cm stratigraphic interval for isothermal remanent magnetization (IRM) measurements. Saturation IRM (SIRM) was imparted in a 2.5 oT field with an impulse magnetizer (model IM-10-30) and was measured with an AGICO JR-6A dual-speed spinner magnetometer in a magnetically shielded laboratory. After SIRM measurement, we further measured backfield IRM imparted at 0.3 T ($IRM_{-300mT}$) by reversing the orientation of samples to calculate the hard IRM (HIRM): $HIRM = (SIRM + IRM_{-300mT})/2$. IRM acquisition curves were also measured at 30 field steps up to 2.7 T for three typical samples.

Manually prepared magnetic extracts were made from five selected samples using a strong rare-earth magnet. FORC measurements of original samples and their magnetic extracts were made using a Princeton Measurements Corporation (Model 3900) vibrating sample magnetometer (VSM). For each sample, 80 FORCs were measured at fields up to ~300 mT, 100 ms averaging time, and 0.3 mT field increment. FORC data were processed using the FORCinel package[68]. Low-temperature magnetic measurements of magnetic extracts were conducted with a Quantum Design superconducting quantum interference device (SQUID) Magnetic Properties Measurement System (MPMS). After cooling in a 5 T field from 300 to 5 K, a low-temperature SIRM, which was imparted in a 5 T field at 5 K, was measured from 5 to 300 K. Then a room temperature SIRM, which was imparted in a 5 T field at 300 K, was measured from 300 to 5 K and back to 300 K in zero field.

**Mineralogical analyses.** Morphological and mineral composition analyses of magnetic extracts were performed using a ZEISS EVO-08 scanning electron microscope (SEM) equipped with a Bruker X-ray energy dispersive spectroscope (EDS). Magnetic extracts were mixed with a low viscosity epoxy and were dispersed by ultrasonication, followed by vacuum-impregnation for 20 min. After curing at 40 °C for 10 h, the specimens were ground with emery paper with successively finer grit size up to 7000 grade and were polished using cloths embedded with 1 μm diamond abrasive. After polishing, the specimens were cleaned ultrasonically in acetone for ~20 min. Finally, the flat polished specimens were coated with a gold layer using an evaporative coater. The polished magnetic extract specimens were then used to determine the morphology of individual particles (including shape and size distributions) and their mineral compositions using the SEM-EDS system[69,70]. Each backscattered electron (BSE) image was obtained with an accelerating voltage of 20 kV and a beam current of 100 μA. The particle size distribution was obtained from 30 BSE pictures from different areas of each polished specimen. An X-ray spectrum was collected for 10 min for each BSE image, and an X-ray dotted map (30 BSE pictures) for each sample was acquired for ~5 h for all chemical elements recognized. X-ray spectra were used to determine the mineral compositions by calculating the net X-ray peak-area of each element[69]. The X-ray maps, with a concentration detection sensitivity of 1‰, were used to detect the spatial mineral distributions.

**Elemental measurements.** Elemental analyses were performed on 175 bulk samples. About 5 g of each sample was dried at 40 °C for 24 h with subsequent grinding to <38 μm (passing a 200-mesh sieve) with an agate mortar and pestle. Powders were then compacted into an oblate polyethylene disc (32-mm diameter) with a tablet machine. The discs were used to determine major element concentrations with an Axios advanced wavelength dispersive X-ray fluorescence instrument (WD-XRF; PANalytical, Ea Almelo, The Netherlands). Relative standard deviations from repeated analyses of the National Standard GSS-8 and GSD-12 were below 2% for all major elements. Grain size analyses were conducted on 768 bulk samples. After removal of organic matter by 10% $H_2O_2$ and carbonate by 10% HCl, samples were measured using a Malvern 2000 laser instrument.

**Astronomical time scale and spectral analyses.** We used an automatic orbital tuning approach[71] to generate an astronomical time scale. We used the 2π-Multi-Taper Method (MTM) to analyze the power spectra with the function Spectral Analysis[72]. Evolutionary power spectra were calculated using the MATLAB Evoffte routine. Both power spectra and evolutionary power spectra were analysed using the Acycle software[73].

A magnetochronology for the Duitinggou section was first established by linear interpolation, based on the ages of the C12r–C12n, C13n–C12r, C13r–C13n, C15n–C13r, C15r–C15n, and C16n.1n–C15r reversal boundaries from the 2012 GPTS[41]. We then conducted spectral analyses of the Lanzhou Basin $\chi_{lf}$ record in the depth domain and using the untuned magnetochronology before tuning, which suggested continuous 405-kyr and 100-kyr eccentricity bands throughout the late Eocene–early Oligocene (Supplementary Figs. 14, 15). Accordingly, we first tuned the 405-kyr component filtered from the $\chi_{lf}$ record to long (405-kyr) eccentricity in the astronomical solution[43] to achieve a cycle-by-cycle correlation within

magnetochronological constraints. Generally, low 405-kyr $\chi_{lf}$ peaks associated with wet climates were tuned to long (405-kyr) eccentricity maxima (Supplementary Fig. 16). We then further refined the age model by fine-adjustment of individual 100-kyr cycles. Low 100-kyr $\chi_{lf}$ peaks (wet climates) were tuned to short (100-kyr) eccentricity maxima within the 405-kyr tuned constraints. To optimize tuning, ages for palaeomagnetic reversals and for 405-kyr tuning were not kept fixed. However, for a few intervals (e.g., ~32.6–33.5 Ma) with stronger 405-kyr variability than 100-kyr variability, the 405-kyr matches were given preference, with simultaneous consideration of resulting sedimentation rate changes, to avoid over-tuning during the 100-kyr tuning stage.

We considered >50 different tuning options for the Duitinggou $\chi_{lf}$ record using age correlation points where high-$\chi_{lf}$ peaks facilitated consistent correlation point selection. Some (~10) of these options resulted in either high 405-kyr correlations but low 100-kyr correlations or high 100-kyr correlations but low 405-kyr correlations within the uncertainty of the magnetochronology. Some (~5) produced both high 405-kyr and 100-kyr correlations, but caused the palaeomagnetic reversal ages to differ too much (up to > 500 kyr) from their GPTS ages. Together, these ~15 options were discarded. In the remaining ~35 options, both 405-kyr and 100-kyr components filtered from the $\chi_{lf}$ record correlate cycle-by-cycle with the target curves, sedimentation rates vary reasonably, and the ages of palaeomagnetic reversals, within uncertainty, are generally consistent with their GPTS ages. All ~35 options produce a similar major spectral evolutionary feature for the $\chi_{lf}$ record, which is characterized by a transition across Oi-1 from dominantly eccentricity to a combination of eccentricity, obliquity, and precession, although the evolutionary intensity and time of each orbital signal vary in different options. From them, we selected the mostly likely option, which contained minimal (42) age correlation points but resulted simultaneously in the high correlation of both the 405-kyr and 100-kyr $\chi_{lf}$ components with their target curves, high consistency of palaeomagnetic reversal age with their GPTS ages, and consistent sedimentation rate changes with lithology (Supplementary Fig. 16; Supplementary Tables 1 and 2). All ~35 options were used to estimate age uncertainties for the tie points. The selected 42 tie points were moved largest forward (older limit) and backward (younger limit) to contain all ~35 possible correlation options to estimate potential positive and negative age uncertainties, respectively.

## Data availability

All data presented this study are accessible openly at the National Tibetan Plateau Data Center (https://data.tpdc.ac.cn/en/data/e3d2b9e4-53d9-4b49-8d9a-9c3d5a7b458c).

## Code availability

Code for orbital tuning and spectral analysis used in this research is available from the corresponding author upon request.

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

## Acknowledgements

This study was supported financially by the Chinese Academy of Sciences (CAS) Strategic Priority Research Program (XDB 40000000), the Second Tibetan Plateau Scientific Expedition and Research (STEP) program (2019QZKK0707, 2019QZKK0101), the CAS Key Research Program of Frontier Sciences (QYZDB-SSW-DQC021), the National Natural Science Foundation of China, the Ministry of Science and Technology of China, Australian Research Council (ARC) Australian Laureate Fellowship grant FL120100050 to E.J.R., ARC grant DP120103952 to A.P.R., ERC consolidor grant MAGIC 649081 to G.D.-N., N.B., N.M., and A.L., and the Bolin Center for Climate Research to H.C.

## Author contributions

H.A., G.D.-N., and Z.S.A. conceived the idea of this study and participated in the fieldwork. E.J.R., A.P.R., A.L., Q.S.L., Z.H.L, and M.J.D. contributed to data analysis, interpretation, and discussion. H.A. and P.Z. performed the magnetic, mineralogical, and multi-proxy measurements. J.-B.L. and C.J.P. contributed to the establishment of related climatic dynamics. Z.H.L., H.K.C., and G.Q.X. helped with marine palaeoclimatic interpretation. Z.D.J., X.K.Q., N.B., N.M., J.Y., and Q.S. helped with terrestrial palaeoclimatic interpretation. C.J.H. helped with the orbital tuning and spectral analysis. H.A., G.D.-N., and Z.S.A. led the manuscript writing with intellectual contributions from all coauthors.

## Competing interests

The authors declare no competing interests.

## Additional information

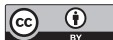

