## [Peer Review File · Nature Communications]

Reviewers' Comments:

Reviewer #1:

Remarks to the Author:

This manuscript presents age model data and multi-proxy paleoenvironmental records from Lanzhou and neighboring areas. The authors found a quite dramatic orbital cycle shift across the Oli-1 event in east Asian continent and they attribute this change to CO₂ and Antarctica ice sheet forcing. I feel that this manuscript is within the scope of the journal, but I have a few suggestions for the authors to consider, some of which are critical to test the validity of their conclusions:

1. The uncertainties associated with age model must be reported in full. I think that both insolation target and geomagnetic reversal boundaries for this old time period have larger uncertainties. Without excluding effects of these uncertainties or considering the uncertainties seriously, it is premature to examine orbital cycles.
2. It is important that the authors show wavelet results of MS using paleomagnetic-based age model. Personally, I am not entirely comfortable with the authors' using tuned age model to talk about eccentricity variations especially the tuning target was eccentricity! This seems like a circular reasoning. So it is important to show the results before tuning and to see if the observed pattern without using the tuned age model still holds. Line 207 only mentions that untuned age model also shows the observed orbital signal shift. This is not enough.
3. How do the authors know that the sequence was continuous, especially before Oli-1 when short orbital cycles were lacking or weak? If sediments before Oli-1 are not continuous or were affected by post-depositional alteration, it can also potentially cause the observed shift in orbital cycles.
4. Why use magnetic susceptibility (MS) to establish the age model? What does MS mean here? Without justification of its climatic implications, I feel that it is slippery to tune it to insolation target. After all, MS is controlled by many factors, such as grain size, concentration, mineralogy, and its variations can be related to provenance, pedogenesis or chemical weathering, etc.
5. I notice that the SAR was generally lower prior to 33.5 Ma. I wonder if the lack of precession and obliquity cycles in the studied stratigraphic records is due to the low SAR and larger effects of post-depositional disturbance? Although MS has a 2.5 kyr sampling resolution of the entire section, I wonder what was the corresponding sampling interval for the interval without precession and obliquity cycles? And how would this larger sampling interval affect the results here?
6. The authors attribute precipitation here to Asian monsoon, but what is the evidence supporting that the moisture during the Eocene was not from Westerlies?
7. The authors argue that eolian input affects one of the proxies (STI), but I wonder if eolian input will also affect the other used proxies here?

Others:

Line 48: asymmetrical seems not the correct word here. Also, I don't buy the temperature reconstruction result here.

179-181: I don't think this record supports your argument here so much. First, it has low resolution, second, it shows great cooling before 33.5 Ma which is lacking for your Hm/Gt record.

184-189: this belongs to discussion

196-200: This is quite a larger change and would require a full discussion and justification of its validity.

229-230: the Xining paleotemperature record has too low resolution to compare with your data here. I feel that neither of these cited records provide solid support to your interpretation of temperature variations in this site. So I suggest you remove this part. It is hard to imagine that Hm/Gt is controlled by temperature with little impact of precipitation. You also mentioned that the benthic oxygen isotope showed a recovery later, which is not consistent with your interpretation of temperature variation here.

232-233: If eolian input affects STI, would it also affect the Hm/Gt record as well? very likely so.

239-240: I don't believe this argument for reasons stated above.

Fig. S3: I feel that the authors need to rigorously discuss the uncertainties associated with their magnetostratigraphy results. The correlation with GPTS is not so straightforward to me. This is particularly the case considering the hot debates about age model for a few sites in China.

Fig. S5: I think in order for the paper to be published, the overprinting should be removed.

Fig. S6: Here MS has a damped response to 400-k forcing during 33-32 Ma, why? Before you figure out the reasoning, I don't feel the tuning has solid foundation.

Fig. S6 caption: I think that the results should be expanded and discussed in detail here.

Reviewer #2:

Remarks to the Author:

Ao et al. present new magnetostratigraphic, rock magnetic, and geochemical data, as well as a new astronomical tuning of the upper Eocene to lower Oligocene sedimentary record from the Lanzhou Basin. These data establish a precise geochronology and record of environmental change for the record. The authors conclude that the onset of substantial hydroclimate change in NE Tibet coincides with Oi-1 of the Eocene-Oligocene Transition. The abstract of the manuscript accurately and clearly represents the principal findings of the work, and I will not repeat them here.

The data are well documented and presented in the manuscript, and the principal conclusions follow directly from these data and the stated assumptions underlying the authors' interpretations. Each figure is essential to the discussion and is clearly illustrated. Many of the interpretations are bolstered by complementary data sets, such as similar chronostratigraphic correlations from neighboring sedimentary sections and independent temperature and precipitation data. The authors present some particularly clear and important links between their Lanzhou Basin record and the regional and global hydrologic cycle (e.g., lines 269..., 290...).

The manuscript is well written, logical, and well referenced. The figures are clear, each is essential to the story, and each is clearly explained. The conclusions will be of interest to a broad variety of Earth System scientists studying the paleogeography and climate of Asia, Cenozoic climate teleconnections, and the Eocene-Oligocene Transition.

I think the manuscript is suitable for publication as it is. I have a few minor suggestions that could clarify the manuscript in a few places, but they are not, in my mind, necessary for publication. I list these below.

Note that I am not able to comment on the details of the astronomical tuning, as this is not my expertise. Hopefully another reviewer can assess this aspect of the work.

Sincerely,

Peter C. Lippert, PhD

Salt Lake City, Utah, USA

17 July 2019

pete.lippert@utah.edu

Minor suggestions/comments:

- a) Inconsistent use of the serial comma in the first few sections of the manuscript;
- b) What about the EOT record from Mongolia: Sun & Windley (2015)?
- c) Line 212: Palike et al. (2012) is another good reference for the Pacific CCD.
- d) Line 227-228: suggestion for more conservative language: ", which we interpret to reflect

temperature changes,...'

e) A statement about the potential for hiatuses, or how you assess the potential for hiatuses in the record, given that the depositional environment (alluvial plain/fan) would be expected to have several. What is the potential for signal shredding (e.g., Jerolmack & Paola, 2010; Wang et al., 2011; Toby et al., 2019) in this record? It appears from the data presented that you have a complete record, but mentioning this potential problem could convince other readers that you're covering all of your potential caveats.

f) Line 299: suggestion: "In summary, we conclude that, following the Oi-1 event,..."

g) Figure 2: your magnetostratigraphic data show a few intervals of positive polarity and less negative polarity within the C12r interval (e.g., 390m, 455m). While this discussion is beyond the scope of this manuscript, given that you have an astrochronology for the record, could these be cryptochrons within C12r (e.g., Cande & Kent, 1992)?

h) In the discussion, the authors jump between reference Pacific and Atlantic records without providing a clear rationale for doing so: I recommend providing some smoother transitions between comparing the various records to minimize the need for readers to read and reread the same section several times to understand why you're talking about an Atlantic record when just previously you were discussing a Pacific record.

i) In Methods and/or Supplement Information, can you please indicate where the various analyses were completed.

j) Line 77 (caption S4) of Supplemental: define NRM: 'NRM labels on the demagnetization diagrams refer to the natural remanent magnetization at room temperature.'

references cited in review:

Cande, S.C., Kent, D.V., 1992. A new geomagnetic polarity timescale for the Late Cretaceous and Cenozoic. *Journal of Geophysical Research* 97, 13917-13951.

Jerolmack, D.J., Paola, C., 2010. Shredding of environmental signals by sediment transport. *Geophysical Research Letters* 37, n/a-n/a, doi:10.1029/2010GL044638.

Pälike, H., Lyle, M.W., Nishi, H., Raffi, I., Ridgwell, A., Gamage, K., Klaus, A., Acton, G., Anderson, L., Backman, J., Baldauf, J., Beltran, C., Bohaty, S.M., BownPaul, Busch, W., Channell, J.E.T., Chun, C.O.J., Delaney, M., Dewangan, P., Dunkley Jones, T., Edgar, K.M., Evans, H., Fitch, P., Foster, G.L., Gussone, N., Hasegawa, H., Hathorne, E.C., Hayashi, H., Herrle, J.O., Holbourn, A., Hovan, S., Hyeong, K., Iijima, K., Ito, T., Kamikuri, S.-i., Kimoto, K., Kuroda, J., Leon-Rodriguez, L., Malinverno, A., Moore Jr, T.C., Murphy, B.H., Murphy, D.P., Nakamura, H., Ogane, K., Ohneiser, C., Richter, C., Robinson, R., Rohling, E.J., Romero, O., Sawada, K., Scher, H., Schneider, L., Sluijs, A., Takata, H., Tian, J., Tsujimoto, A., Wade, B.S., Westerhold, T., Wilkens, R., Williams, T., Wilson, P.A., Yamamoto, Y., Yamamoto, S., Yamazaki, T., Zeebe, R.E., 2012. A Cenozoic record of the equatorial Pacific carbonate compensation depth. *Nature* 488, 609-614, doi:10.1038/nature11360.

Toby, S.C., Duller, R.A., De Angelis, S., Straub, K.M., 2019. A Stratigraphic Framework for the Preservation and Shredding of Environmental Signals. *Geophysical Research Letters* 46, 5837-5845, doi:10.1029/2019gl082555.

Sun, J., Windley, B.F., 2015. Onset of aridification by 34 Ma across the Eocene-Oligocene transition in Central Asia. *Geology* 43, 1015-1018, doi:10.1130/g37165.1.

Wang, Y., Straub, K.M., Hajek, E.A., 2011. Scale-dependent compensational stacking: An estimate of autogenic time scales in channelized sedimentary deposits. *Geology* 39, 811-814, doi:10.1130/g32068.1.

Reviewer #3:

Remarks to the Author:

The Eocene–Oligocene climate transition (EOT) have been well recorded by marine deposits, during which global temperature stepwise decreased. High resolution records in continents are compared with marine climate changes during EOT would be helpful to understand dynamics of climate system. This kind of terrestrial EOT records are generally rare, in particular in East Asia, except studies from Xining Basin in the NE Tibetan Plateau. In this manuscript, Ao et al. have provided high-resolution records of paleomagnetic stratigraphy and environmental magnetism from Lanzhou Basin, ~200 km east to Xining Basin, to document the EOT climate change on Northern Tibetan Plateau. They draw two main conclusions: (1) rapid aridification and onset of cooling occurred at Oi-1 (~33.7 Ma); (2) the hydroclimate oscillations shifted eccentricity cycles to a combination of eccentricity, obliquity, and precession immediately after Oi-1. The idea about the regional climate change during EOT is not fully new, similar conclusions have been reported in the Xining Basin (Dupont-Nivet et al., 2007; Fang et al., 2019; Fang et al., 2015; Page et al., 2019; Xiao et al., 2010; Zhang and Guo, 2014).

My major concern on this study is regarding the reliability about their geochemical proxies, including the Hm/Gt and Silica-Titania Index (STI). These proxies are neither well accepted nor well calibrated in the literature or the authors' manuscript. Hm/Gt was rarely, if not never, used as a simple temperature proxy before. The Hm/Gt ratio changes have been suggested as a change in the timing of precipitation, a change in temperature, or some combination of these factors in the Luochuan and Lingtai loess sections (Balsam et al., 2004), while others use the Hm/Gt ratio as a proxy of dry/humid variability (Ji et al., 2004; Zhang et al., 2007). Unfortunately, the authors interpreted the Hm/Gt ratio of fluvial and lacustrine sediments in Lanzhou Basin as regional temperature changes just because "precipitation increase is unlikely because of evidence provided here and elsewhere (7, 9-11, 22) that indicates enhanced aridification and lower precipitation supply across the EOT in this area" (Line 174-175). Moreover, the authors should also be aware that Gt is not a stable mineral and it is hard to imagine that initial Hm/Gt could be faithfully preserved for tens of millions of years in the detrital sediment basin. The authors must therefore provide more convincing discussions on why they can use Hm/Gt as a temperature proxy. It is also weird to use Silica-Titania Index as a weathering intensity proxy as very few readers including me really know this proxy. I'm wondering why CIA or other conventional elemental weathering proxies were not used? At the current stage, it gives me an impression that the authors were simply trying to fit proxy records into the story of EOT cooling and aridification without properly consider the suitability of those proxies.

Overall, I consider this study could be a useful contribution of new data to understand the early Cenozoic paleoclimate change in NW China and can be accepted for publication when my comments are addressed in their revision.

More specific comments below:

1. The geological background of their study region, i.e. Lanzhou Basin is missing. Without an introduction of tectonic and sedimentary context, the readers have no trace of where the sediments come from, and whether the sediments can record the regional environment changes during EOT. In line 73-78, the authors cited many works on the Xining Basin, which do not back up the feasibility of paleoclimate reconstruction on sediments from Lanzhou Basin. The authors should make it clear on the current understanding of EOT climate and environment changes in the studied region and what are the new findings in this study?

2. The paleomagnetic stratigraphy analysis of this study is based on several previous studies (Yue et

al., 2001; Zhang et al., 2016; Zhang et al., 2018) in particular based on Yue et al. (2001). Is there any significantly chronological improvements the authors have done in this study? Or, it is just a high-resolution sampling and analysis?

3. What controlled the variation of magnetic properties such as magnetic susceptibility (MS), saturation isothermal remanent magnetization (SIRM), hard isothermal remnant magnetization (HIRM) of the studied depositional sequence? Without such information in the manuscript, the orbital cycle analysis on this proxy are less convincing. I don't think the record resolution is high enough to study the precession cycles.

4. The authors used the glacial-interglacial alternations model to interpret the paleoclimate change in the EOT (lines 293-298), during which the environment background is totally different from that of the Pleistocene glaciations. Are they really comparable?

In Asia, the EOT cooling has been documented, but the wet/dry change has not been well resolved. I hope this paper would contribute to these important questions in monsoonal Asia, rather than just confirm previous conclusions. I would say, there is a great space to improve the quality of this paper. I would suggest the authors to revise their paper substantially and I am happy to review the revised manuscript again.

References

- Balsam, W., Ji, J., Chen, J., 2004. Climatic interpretation of the Luochuan and Lingtai loess sections, China, based on changing iron oxide mineralogy and magnetic susceptibility. *Earth Planet Sc Lett* 223, 335-348.
- Dupont-Nivet, G., Krijgsman, W., Langereis, C.G., Abels, H.A., Dai, S., Fang, X., 2007. Tibetan plateau aridification linked to global cooling at the Eocene-Oligocene transition. *Nature* 445, 635-638.
- Fang, X., Fang, Y., Zan, J., Zhang, W., Song, C., Appel, E., Meng, Q., Miao, Y., Dai, S., Lu, Y., Zhang, T., 2019. Cenozoic magnetostratigraphy of the Xining Basin, NE Tibetan Plateau, and its constraints on paleontological, sedimentological and tectonomorphological evolution. *Earth-Sci Rev* 190, 460-485.
- Fang, X., Zan, J., Appel, E., Lu, Y., Song, C., Dai, S., Tuo, S., 2015. An Eocene-Miocene continuous rock magnetic record from the sediments in the Xining Basin, NW China: indication for Cenozoic persistent drying driven by global cooling and Tibetan Plateau uplift. *Geophysical Journal International* 201, 78-89.
- Ji, J., Chen, J., Balsam, W., Lu, H., Sun, Y., Xu, H., 2004. High resolution hematite/goethite records from Chinese loess sequences for the last glacial-interglacial cycle: Rapid climatic response of the East Asian Monsoon to the tropical Pacific. *Geophysical Research Letters* 31.
- Page, M., Licht, A., Dupont-Nivet, G., Meijer, N., Barbolini, N., Hoorn, C., Schauer, A., Huntington, K., Bajnai, D., Fiebig, J., Mulch, A., Guo, Z., 2019. Synchronous cooling and decline in monsoonal rainfall in northeastern Tibet during the fall into the Oligocene icehouse. *Geology* 47, 203-206.
- Xiao, G., Abels, H.A., Yao, Z., Dupont-Nivet, G., Hilgen, F.J., 2010. Asian aridification linked to the first step of the Eocene-Oligocene climate Transition (EOT) in obliquity-dominated terrestrial records (Xining Basin, China). *Climate of the Past* 6, 501.
- Yue, L.P., Heller, F., Qiu, Z.X., Zhang, L., Xie, G.P., Qiu, Z.D., Zhang, Y.X., 2001. Magnetostratigraphy

and paleoenvironmental record of Tertiary deposits of Lanzhou Basin. *Chinese Sci Bull* 46, 770-774.

Zhang, C., Guo, Z., 2014. Clay mineral changes across the Eocene–Oligocene transition in the sedimentary sequence at Xining occurred prior to global cooling. *Palaeogeography, Palaeoclimatology, Palaeoecology* 411, 18-29.

Zhang, P., Ao, H., Dekkers, M.J., An, Z., Wang, L., Li, Y., Liu, S., Qiang, X., Chang, H., Zhao, H., 2018. Magnetostratigraphy of the Oligocene mammalian faunas in the Lanzhou Basin, Northwest China. *Journal of Asian Earth Sciences* 159, 24-33.

Zhang, P., Ao, H., Dekkers, M.J., Li, Y., An, Z., 2016. Late Oligocene-Early Miocene magnetostratigraphy of the mammalian faunas in the Lanzhou Basin-environmental changes in the NE margin of the Tibetan Plateau. *Scientific Reports* 6.

Zhang, Y.G., Ji, J., Balsam, W.L., Liu, L., Chen, J., 2007. High resolution hematite and goethite records from ODP 1143, South China Sea: Co-evolution of monsoonal precipitation and El Niño over the past 600,000 years. *Earth Planet Sc Lett* 264, 136-150.

Reply to Reviewer #1

Comment #1

The uncertainties associated with age model must be reported in full. I think that both insolation target and geomagnetic reversal boundaries for this old time period have larger uncertainties. Without excluding effects of these uncertainties or considering the uncertainties seriously, it is premature to examine orbital cycles.

Response: We have added discussion on age model uncertainties to the Supplementary Information (Supplementary Note 3). Between 36 and 30 Ma, the La₂₀₀₄ (Laskar et al., 2004) solution curves comes with small uncertainties, but the updated La₂₀₁₀ solutions (Laskar et al., 2011) used here have only minute uncertainty (see Figure 1 below). At present, large uncertainties associated with insolation calculations exist primarily before the Cenozoic (Laskar et al., 2011), which is beyond the time interval as studied here.

Compared with the solutions of insolation time series, age uncertainties for the geomagnetic reversal boundaries are somewhat larger, but are still generally < 400 kyr (Table S2), i.e., less than a long eccentricity cycle, the starting point of our tuning exercise. Long 405-kyr eccentricity cycles are the most stable and characteristic climate rhythm during the Late Eocene and Early Oligocene (Coxall et al., 2005; De Vleeschouwer et al., 2017; Levy et al., 2019; Pälike et al., 2006). Thus, our first-order tuning of 405-kyr eccentricity cycles provides a reasonably refined age model under the magnetochronological constraints. Further tuning of 100-kyr eccentricity cycles is done routinely to further refine age models. Uncertainties associated with our combined magneto- and astro-chronology, thus, decrease significantly to small levels. Certain short intervals may occasionally have larger uncertainties, but they would not influence the observed marked orbital shift across the Oi-1 event in the astronomically tuned Lanzhou χ record (Fig. 4), which is deemed robust. The untuned magnetochronology clearly has similar orbital variability (Supplementary Fig. 12), which is also captured by the global marine benthic $\delta^{18}\text{O}$ climatic proxy (Fig. 4). These results all support the insignificant effects of uncertainties on our novel finding of the orbital shift at 33.65 Ma. Therefore, our new astronomical chronology based on 405- and 100-kyr tuning refines the age model, decreases age uncertainties, and improves the orbital χ expression (eccentricity, obliquity, and precession).

Figure 1. Mean daily insolation at 65°N for La₂₀₀₄ (Laskar et al., 2004) and four La₂₀₁₀ solutions (Laskar et al., 2011) between 36 and 30 Ma. La₂₀₀₄ solution is in phase with La₂₀₁₀ solutions and is accurate for ages less than 32 Ma. All the four La₂₀₁₀ solutions, which have higher precision than La₂₀₀₄ solution, are almost identical, as shown (they are essentially on top of each other), and are accurate over the past 36 Ma.

Coxall, H.K., Wilson, P.A., Pälike, H., Lear, C.H., Backman, J., 2005. Rapid stepwise onset of Antarctic glaciation and deeper calcite compensation in the Pacific Ocean. *Nature* 433, 53–57.

De Vleeschouwer, D., Vahlenkamp, M., Crucifix, M., Pälike, H., 2017. Alternating Southern and Northern Hemisphere climate response to astronomical forcing during the past 35 m.y. *Geology* 45, 375–378.

Laskar, J., Fienga, A., Gastineau, M., Manche, H., 2011. La₂₀₁₀: a new orbital solution for the long-term motion of the Earth. *Astron. Astrophys.* 532, A89, doi: 10.1051/0004-6361/201116836

Laskar, J., Robutel, P., Joutel, F., Gastineau, M., Correia, A.C.M., Levrard, B., 2004. A long-term numerical solution for the insolation quantities of the Earth. *Astron. Astrophys.* 428, 261–285.

Levy, R.H., Meyers, S.R., Naish, T.R., Golledge, R.N., McKay, R.M., Crampton, J.S., DeConto, R.M., De Santis, L., Florindo, F., Gasson, E.G.W., Harwood, D.M., Luyendyk, B.P., Powell, R.D., Clowes, C., Kulhanek, D.K., 2019. Antarctic ice-sheet sensitivity to obliquity forcing enhanced through ocean connections. *Nat. Geosci.* 12, 132–137.

Pälike, H., Norris, R.D., Herrle, J.O., Wilson, P.A., Coxall, H.K., Lear, C.H., Shackleton, N.J., Tripathi, A.K., Wade, B.S., 2006. The heartbeat of the Oligocene climate system. *Science* 314, 1894–1898.

Comment #2

It is important that the authors show wavelet results of MS using paleomagnetic-based age model. Personally, I am not entirely comfortable with the authors' using tuned age model to talk about eccentricity variations especially the tuning target was eccentricity! This seems like a circular reasoning. So it is important to show the results before tuning and to see if the observed pattern without using the tuned age model still holds. Line 207 only mentions that untuned age model also shows the observed orbital signal shift. This is not enough.

Response: We agree that it is important to show the orbital periodicity evolution of χ on the untuned

magnetostratigraphy. It is now shown in Supplementary Figure 12. The major orbital shift across Oi-1 in our astronomical timescale is also captured in the untuned magnetostratigraphy. This strengthens our conclusions and indicates that this result is not artifact of the orbital tuning procedure.

Comment #3

How do the authors know that the sequence was continuous, especially before Oli-1 when short orbital cycles were lacking or weak? If sediments before Oli-1 are not continuous or were affected by post-depositional alteration, it can also potentially cause the observed shift in orbital cycles.

Response: This is a good point. As acknowledged above and in the revised manuscript, potential unidentified small sedimentary hiatuses or post-depositional alternations may have caused one or several obliquity or precession cycles to be missing over a few comparatively short intervals, although such disruptions are not evident in the Late Eocene–Oligocene Duitingou section. However, we argue that the observed orbital cyclicity shift characterized by obliquity and precession absence before 33.65 and their later appearance remains robust, and is highly unlikely to result from a few undetectable small gaps that cannot have caused essentially all obliquity and precession cycles to disappear over the ~1.91 Myr long 35.56–33.65 Ma interval.

Comment #4

Why use magnetic susceptibility (MS) to establish the age model? What does MS mean here? Without justification of its climatic implications, I feel that it is slippery to tune it to insolation target. After all, MS is controlled by many factors, such as grain size, concentration, mineralogy, and its variations can be related to provenance, pedogenesis or chemical weathering, etc.

Response: Magnetic susceptibility (χ) is a proxy for sedimentary magnetic mineral concentration. We conducted systematic rock magnetic analyses and now provided a more in-depth discussion to further determine the mechanism for observed χ variations and associated linkage with palaeoclimate variability (lines 162–183; Supplementary Figs. 7–10). Positive linear correlations of χ , SIRM, and HIRM (Supplementary Fig. 8) together with their consistent orbital oscillations (Supplementary Fig. 9) all reflect magnetic mineral concentration changes. Magnetic mineral concentration variations have been shown to be related to palaeoclimatic changes in Xining Basin studies in combination with sedimentary, clay mineral, and pollen data (Dupont-Nivet et al., 2007; Fang et al., 2015; Page et al.,

2019; Xiao et al., 2010a, 2010b; Zhang and Guo, 2014). Both Lanzhou and Xining basins were part of the larger Longzhong Basin and were influenced by the same Palaeogene hydroclimate. Thus, we use our χ record, which has the highest sampling resolution of the magnetic parameters measured in our study, as a palaeoclimate proxy to establish the astronomical chronology.

Dupont-Nivet, G., Krijgsman, W., Langereis, C.G., Abels, H.A., Dai, S., Fang, X.M., 2007. Tibetan Plateau aridification linked to global cooling at the Eocene-Oligocene transition. *Nature* 445, 635-638.

Fang, X.M., Zan, J.B., Appel, E., Lu, Y., Song, C.H., Dai, S., Tuo, S.B., 2015. An Eocene-Miocene continuous rock magnetic record from the sediments in the Xining Basin, NW China: indication for Cenozoic persistent drying driven by global cooling and Tibetan Plateau uplift. *Geophysical Journal International* 201, 78-89.

Page, M., Licht, A., Dupont-Nivet, G., Meijer, N., Barbolini, N., Hoorn, C., Schauer, A., Huntington, K., Bajnai, D., Fiebig, J., Mulch, A., Guo, Z., 2019. Synchronous cooling and decline in monsoonal rainfall in northeastern Tibet during the fall into the Oligocene icehouse. *Geology* 47, 203-206.

Xiao, G.Q., Abels, H.A., Yao, Z.Q., Dupont-Nivet, G., Hilgen, F.J., 2010a. Asian aridification linked to the first step of the Eocene-Oligocene climate Transition (EOT) in obliquity-dominated terrestrial records (Xining Basin, China). *Climate of the Past* 6, 501-513.

Xiao, G.Q., Zhou, X.Y., Ge, J.Y., Zhan, T., Yao, Z.Q., 2010b. Sedimentary characteristics and paleoenvironmental significance of late Eocene gypsum-mudston cycles in the Xining Basin, northeastern Tibetan Plateau. *Quaternary Sciences* 30, 919-924.

Zhang, C.X., Guo, Z.T., 2014. Clay mineral changes across the Eocene-Oligocene transition in the sedimentary sequence at Xining occurred prior to global cooling. *Palaeogeography Palaeoclimatology Palaeoecology* 411, 18-29

Comment #5

I notice that the SAR was generally lower prior to 33.5 Ma. I wonder if the lack of precession and obliquity cycles in the studied stratigraphic records is due to the low SAR and larger effects of post-depositional disturbance? Although MS has a 2.5 kyr sampling resolution of the entire section, I

wonder what was the corresponding sampling interval for the interval without precession and obliquity cycles? And how would this larger sampling interval affect the results here?

Response: We agree that explaining this point in more detail strengthens our argumentation. The sampling interval for χ measurements is consistently 10 cm through the Late Eocene–Oligocene Duitinggou section. We now additionally discuss potential impacts of sampling resolution and disruptions on recording of orbital variations (lines 192–196 and 249–257). Sedimentation rates are relatively lower prior to 33.65 Ma, but the associated χ resolution is still high and varies between 2 and 4 kyr, which allows meaningful resolution of obliquity and precession cycles if they are present. In particular, in the younger portion of the record, after 33.65 Ma, with essentially the same sampling resolution (1–3.5 Kyr), strong precession cycles are observed. Thus, sedimentation rate issues and sampling resolution cannot be the reason for the absence of obliquity and precession cycles throughout the long 35.56–33.65 Ma portion of the record.

Comment #6

The authors attribute precipitation here to Asian monsoon, but what is the evidence supporting that the moisture during the Eocene was not from Westerlies?

Response: Yes, both Asian monsoon and westerlies are involved when discussing Lanzhou Basin precipitation transportation. We have revised the manuscript to avoid such misleading statements.

Comment #7

The authors argue that eolian input affects one of the proxies (STI), but I wonder if eolian input will also affect the other used proxies here?

Response: Thanks for pointing out this. In addition to a major decrease in regional chemical weathering intensity, increased aeolian input from drier regions in the north and west of Lanzhou Basin with weaker chemical weathering may have additionally amplified the CIA, K_2O/Al_2O_3 , and STI shifts to produce dramatic changes across the Oi-1 event. We have added this discussion (lines 215-217). The magnetic mineral concentration of fluvial-lacustrine sediments in the Lanzhou Basin is dominated by detrital input due to rainfall and river runoff, while the contribution from aeolian input is small. Increased weakly magnetic aeolian input from drier regions in the north and west of Lanzhou Basin may have influenced slightly the mean trends of magnetic proxies before and after Oi-1, but not their

orbital variability, which is the focus of our study and remains robust.

Comment #8

Line 48: asymmetrical seems not the correct word here. Also, I don't buy the temperature reconstruction result here.

Response: We appreciate reviewers 1 and 3 for pointing out a complicated process of fluvial-lacustrine Hm/Gt variability in the > 30 Ma sediments. We agree that the presently available results are not sufficient to document a robust temperature interpretation of the Hm/Gt shift across Oi-1. Thus, we removed the temperature discussion from Hm/Gt as suggested, and no longer discuss asymmetry between precipitation and temperature.

Comment #9

179-181: I don't think this record supports your argument here so much. First, it has low resolution, second, it shows great cooling before 33.5 Ma which is lacking for your Hm/Gt record.

Response: We removed the Hm/Gt record and discussion, so we no longer use the clumped isotope record from the neighbouring Xining Basin to support the Hm/Gt temperature reconstruction.

Comment #10

184-189: this belongs to discussion.

Response: We agree that it is generally better to discuss results in a discussion section. However, after moving the text from the data section to the discussion section, we find that it fits better in the former section. This allows readers to more easily and immediately understand the two factors (weathering intensity and aeolian input) that influence the CIA, K_2O/Al_2O_3 , and STI shifts. When this text about aeolian input influence is moved to the discussion, questions concerning other causes to explain the chemical weathering proxy shifts remain unaddressed until aeolian inputs are discussed much later, which results in a logic gap.

Comment #11

196-200: This is quite a larger change and would require a full discussion and justification of its

validity.

Response: We now provide a more extended justification (lines 228-235). The origin of such a large change is now clearer.

Comment #12

229-230: the Xining paleotemperature record has too low resolution to compare with your data here. I feel that neither of these cited records provide solid support to your interpretation of temperature variations in this site. So I suggest you remove this part. It is hard to imagine that Hm/Gt is controlled by temperature with little impact of precipitation. You also mentioned that the benthic oxygen isotope showed a recovery later, which is not consistent with your interpretation of temperature variation here.

Response: We are thankful for this insightful suggestion. We have removed the Hm/Gt temperature reconstruction as suggested. The low-resolution Xining Basin clumped isotope record is mentioned to support Asian cooling across the EOT but it has been removed from Figure 3 because it does not make sense to compare it with our high-resolution records.

Comment #13

232-233: If eolian input affects STI, would it also affect the Hm/Gt record as well? very likely so.

Response: The Hm/Gt record and discussion have been removed as suggested.

Comment #14

239-240: I don't believe this argument for reasons stated above.

Response: Hm/Gt-related discussions have been removed as suggested.

Comment #15

Fig. S3: I feel that the authors need to rigorously discuss the uncertainties associated with their magnetostratigraphy results. The correlation with GPTS is not so straightforward to me. This is particularly the case considering the hot debates about age model for a few sites in China.

Response: Few magnetostratigraphic records for older Cenozoic strata in western China are reliable over the entire section, which generally contain some inconsistent correlations with the GPTS, but the

Duitinggou and neighbouring Xining Basin records are rare exceptions. We have added further discussion to clarify uncertainties in our magnetostratigraphic results and to highlight the reliability of the correlation (Supplementary Note 2). Our magnetostratigraphic results also incorporate published data that constrain the ages above and below our record as indicated in Figures 2 and Supplementary Figure 4.

Comment #16

Fig. S5: I think in order for the paper to be published, the overprinting should be removed.

Response: In the revised manuscript, we combine our previously published 614 ChRM directions from the overlying Xianshuihe Formation with our 295 new directions from the Yehucheng Formation (from the same Duitinggou section) to provide sufficient palaeomagnetic data for an adequate reversals test with improved statistical significance. Our new results have improved clustering with antipodal normal and reversed polarity groups (Supplementary Fig. 6) and pass a class C reversals test of McFadden and McElhinny (1990). This indicates that the Duitinggou magnetostratigraphy is reliable, with clear and unequivocal correlation to the GPTS (Fig. 2). Potential overprint is minor, if present at all, and does not affect overall determination of normal and reversed polarity zones and their correlation to the GPTS. The reversals test is now positive: our previous interpretation of overprinting to explain the failed reversals test is no longer valid. It was based on a much shorter time span with a smaller sample set.

Comment #17

Fig. S6: Here MS has a damped response to 405-k forcing during 33-32 Ma, why? Before you figure out the reasoning, I don't feel the tuning has solid foundation.

Response: The attenuated response is consistent with the coeval enhanced response to 100-kyr eccentricity forcing (Fig. 4); the 100-kyr eccentricity response became much higher than the 405-kyr eccentricity response between 32.7 and 31.9 Ma.

Comment #18

Fig. S6 caption: I think that the results should be expanded and discussed in detail here.

Response: Good suggestion. It is important to clarify these issues; detailed discussion on astronomical

timescale construction with associated uncertainties has been added to the Supplementary Information (Supplementary Note 3).

Reply to Reviewer #2

Comment #1

a) Inconsistent use of the serial comma in the first few sections of the manuscript.

Response: We have carefully checked this.

Comment #2

b) What about the EOT record from Mongolia: Sun & Windley (2015)?

Response: Thanks for reminding us the study of Sun and Windley (2015). Records from Mongolia (Kraatz and Geisler, 2010; Sun and Windley, 2015) and Junggar Basin (Sun et al., 2014) also suggest a general Asian aridification trend across the EOT. We have cited them to provide additional support for Asian aridification across the EOT (line 67).

Comment #3

c) Line 212: Pälke et al. (2012) is another good reference for the Pacific CCD.

Response: This reference has been added.

Comment #4

d) Line 227-228: suggestion for more conservative language: “, which we interpret to reflect temperature changes, ...”

Response: We have removed discussion of Hm/Gt in relation to temperature changes in response to Reviewer #1.

Comment #5

e) A statement about the potential for hiatuses, or how you assess the potential for hiatuses in the record, given that the depositional environment (alluvial plain/fan) would be expected to have several. What

is the potential for signal shredding (e.g., Jerolmack & Paola, 2010; Wang et al., 2011; Toby et al., 2019) in this record? It appears from the data presented that you have a complete record, but mentioning this potential problem could convince other readers that you're covering all of your potential caveats.

Response: Thank you for pointing this out. We have added discussion of this issue to the Supplementary Information (Supplementary Note 3). See also comment #3 from reviewer 1.

Comment #6

f) Line 299: suggestion: "In summary, we conclude that, following the Oi-1 event,...".

Response: Done.

Comment #7

g) Figure 2: your magnetostratigraphic data show a few intervals of positive polarity and less negative polarity within the C12r interval (e.g., 390m, 455m). While this discussion is beyond the scope of this manuscript, given that you have an astrochronology for the record, could these be cryptochrons within C12r (e.g., Cande & Kent, 1992)?

Response: A few intervals with a single positive direction within C12r may indicate short-lived palaeomagnetic anomalies (or tiny wiggles) with an ambiguous origin or excursions (or cryptochrons) that need to be documented worldwide before they can be considered 'real'. In this study, each polarity zone is defined by at least three successive levels with the same polarity in an attempt to avoid interpretation of geomagnetic excursions as polarity chrons or subchrons. Related discussion has been added to the Supplementary Information (Supplementary Note 2).

Comment #8

h) In the discussion, the authors jump between reference Pacific and Atlantic records without providing a clear rationale for doing so: I recommend providing some smoother transitions between comparing the various records to minimize the need for readers to read and reread the same section several times to understand why you're talking about an Atlantic record when just previously you were discussing a Pacific record.

Response: Thank you. The logic is now improved. In our terrestrial-to-marine comparisons of orbital variability, we now focus on the Pacific Ocean benthic $\delta^{18}\text{O}$ record, which is presently the highest-resolution available. Both Pacific and Atlantic records are used for long-term terrestrial-to-marine correlations, but the jumping back and forth in the logic has been removed.

Comment #9

i) In Methods and/or Supplementary Information, can you please indicate where the various analyses were completed.

Response: This has been added to the “Supplementary Note 1” in the Supplementary Information.

Comment #10

j) Line 77 (caption S4) of Supplemental: define NRM: ‘NRM labels on the demagnetization diagrams refer to the natural remanent magnetization at room temperature.’

Response: Done (see caption to Supplementary Fig. 5).

Reply to Reviewer #3

Comment #1

The Eocene–Oligocene climate transition (EOT) have been well recorded by marine deposits, during which global temperature stepwise decreased. High resolution records in continents are compared with marine climate changes during EOT would be helpful to understand dynamics of climate system. This kind of terrestrial EOT records are generally rare, in particular in East Asia, except studies from Xining Basin in the NE Tibetan Plateau. In this manuscript, Ao et al. have provided high-resolution records of paleomagnetic stratigraphy and environmental magnetism from Lanzhou Basin, ~200 km east to Xining Basin, to document the EOT climate change on Northern Tibetan Plateau. They draw two main conclusions: (1) rapid aridification and onset of cooling occurred at Oi-1 (~33.7 Ma); (2) the hydroclimate oscillations shifted eccentricity cycles to a combination of eccentricity, obliquity, and precession immediately after Oi-1. The idea about the regional climate change during EOT is not fully new, similar conclusions have been reported in the Xining Basin (Dupont-Nivet et al., 2007; Fang et al., 2019; Fang et al., 2015; Page et al., 2019; Xiao et al., 2010; Zhang and Guo, 2014).

Response: Yes, Asian cooling and aridification across the EOT has been inferred from Xining, Maoming, and Junggar basins, and Mongolia. These results are now more cited. We also clearly indicate in the introduction that our novel finding of orbital shift across the Oi-1 and established forcing dynamics has not been reported in previous studies and is an entirely new contribution to EOT studies.

Comment #2

My major concern on this study is regarding the reliability about their geochemical proxies, including the Hm/Gt and Silica-Titania Index (STI). These proxies are neither well accepted nor well calibrated in the literature or the authors' manuscript. Hm/Gt was rarely, if not never, used as a simple temperature proxy before. The Hm/Gt ratio changes have been suggested as a change in the timing of precipitation, a change in temperature, or some combination of these factors in the Luochuan and Lingtai loess sections (Balsam et al., 2004), while others use the Hm/Gt ratio as a proxy of dry/humid variability (Ji et al., 2004; Zhang et al., 2007). Unfortunately, the authors interpreted the Hm/Gt ratio of fluvial and lacustrine sediments in Lanzhou Basin as regional temperature changes just because "precipitation increase is unlikely because of evidence provided here and elsewhere (7, 9-11, 22) that indicates enhanced aridification and lower precipitation supply across the EOT in this area" (Line 174-175). Moreover, the authors should also be aware that Gt is not a stable mineral and it is hard to imagine that initial Hm/Gt could be faithfully preserved for tens of millions of years in the detrital sediment basin. The authors must therefore provide more convincing discussions on why they can use Hm/Gt as a temperature proxy. It is also weird to use Silica-Titania Index as a weathering intensity proxy as very few readers including me really know this proxy. I'm wondering why CIA or other conventional elemental weathering proxies were not used? At the current stage, it gives me an impression that the authors were simply trying to fit proxy records into the story of EOT cooling and aridification without properly consider the suitability of those proxies.

Response: We agree and are grateful for this insightful comment (except the part on trying to fit the proxy record!), which has strengthened our study significantly. Both reviewers 1 and 3 have similar comments on complex Hm/Gt variability in these old fluvial-lacustrine sediments. We now agree that the Hm/Gt data are not sufficient for robust temperature interpretation, so we have removed this material altogether, as also suggested by Reviewer 1. This deletion does not influence our novel finding of the orbital shift across the Oi-1, which is based on the high-resolution magnetic susceptibility record.

In addition to STI, we have added CIA and K_2O/Al_2O_3 records, as suggested. They have the same variability and suggest consistently decreased chemical weathering intensity across the Oi-1 event.

Comment #3

The geological background of their study region, i.e. Lanzhou Basin is missing. Without an introduction of tectonic and sedimentary context, the readers have no trace of where the sediments come from, and whether the sediments can record the regional environment changes during EOT. In line 73-78, the authors cited many works on the Xining Basin, which do not back up the feasibility of paleoclimate reconstruction on sediments from Lanzhou Basin. The authors should make it clear on the current understanding of EOT climate and environment changes in the studied region and what are the new findings in this study?

Response: Thank you for this comment. Several of our previous publications cover these background aspects extensively for the Lanzhou Basin and we apologize for not including them sufficiently here. Geological background information for Lanzhou Basin has been added (lines 102–111). In addition to discussing marine and terrestrial EOT trends, we also summarize our current understanding of global and terrestrial EOT climate and environmental changes in the introduction. We hope our new findings are also clearer with changes to lines 81-95.

Comment #4

The paleomagnetic stratigraphy analysis of this study is based on several previous studies (Yue et al., 2001; Zhang et al., 2016; Zhang et al., 2018) in particular based on Yue et al. (2001). Is there any significantly chronological improvements the authors have done in this study? Or, it is just a high-resolution sampling and analysis?

Response: Our previous studies (Zhang et al., 2016, 2018) established a magnetostratigraphy (~18–31 Ma) for the Xianshuihe Formation at the Duitinggou section (0–383 m), which indeed provides a basis to establish a magnetostratigraphy for the concordantly underlying (older) Yehucheng Formation (383–570 m). Yue et al. (2001) provided a major record to support the robustness of our magnetostratigraphy. Our new age model has significant improvements although the pioneering magnetostratigraphic work of Yue et al. (2001) was not far off. In addition to higher-resolution sampling and more detailed analysis, we extended the record in time and space in various sections. The reliability of the

magnetostratigraphic correlation is therefore improved. Another significant chronological improvement is that we establish a more precise astronomical timescale by orbital tuning using the magnetostratigraphy.

Comment #5

What controlled the variation of magnetic properties such as magnetic susceptibility (MS), saturation isothermal remanent magnetization (SIRM), hard isothermal remnant magnetization (HIRM) of the studied depositional sequence? Without such information in the manuscript, the orbital cycle analysis on this proxy are less convincing. I don't think the record resolution is high enough to study the precession cycles.

Response: Excellent point. The origin of χ , SIRM, and HIRM signals and their links to palaeoclimate variability are now discussed more extensively and justified with additional data from rock magnetic analyses and comparison to other proxies and records (lines 162–184; Supplementary Figs. 7–10). The sample resolution issue is now examined in more detail to demonstrate that it is sufficient to resolve precession cycles. So, this is not the reason why precession and obliquity are absent throughout the long-term 35.56–33.65 Ma interval, as is now stated clearly (lines 249-257). We have also toned down the discussion of a precession forcing shift across Oi-1 in the Lanzhou χ record and agree that obtaining higher-resolution records would be ideal to make a solid case for shifting precession forcing.

Comment #6

The authors used the glacial-interglacial alternations model to interpret the paleoclimate change in the EOT (lines 293-298), during which the environment background is totally different from that of the Pleistocene glaciations. Are they really comparable?

Response: We agree that Oligocene “glacial-interglacial” alternations were dominated by Antarctic ice-sheet advances and retreats. The associated environmental background differs substantially from Quaternary glacial-interglacial alternations, which are dominated by both Antarctic and high-latitude northern hemisphere ice sheets (Raymo et al., 2006). However, we argue that some general large-scale climate dynamics should remain grossly similar. Oligocene glacials had lower sea level, CO₂, and temperatures than interglacials (Miller et al., 2009; Pälike et al., 2006). Thus, it seems more likely that these large-scale Oligocene climatic processes and glacial-interglacial alternations played an important

role in orbital Asian climate variability. We have toned down the discussion and added a note to the revised manuscript (lines 307-324) that caution should be used to make inferences about environmental background differences.

Miller, K.G., Wright, J.D., Katz, M.E., Wade, B.S., Browning, J.V., Cramer, B.S., Rosenthal, Y., 2009.

Climate threshold at the Eocene-Oligocene transition: Antarctic ice sheet influence on ocean circulation. *Geological Society of America Special Papers* 452, 169-178.

Pälike, H., Norris, R.D., Herrle, J.O., Wilson, P.A., Coxall, H.K., Lear, C.H., Shackleton, N.J., Tripathi,

A.K., Wade, B.S., 2006. The heartbeat of the Oligocene climate system. *Science* 314, 1894-1898.

Raymo, M.E., Lisiecki, L.E., Nisancioglu, K.H., 2006. Plio-Pleistocene ice volume, Antarctic climate, and the global $\delta^{18}\text{O}$ record. *Science* 313, 492-495.

Reviewers' Comments:

Reviewer #1:

Remarks to the Author:

This manuscript has been clearly improved. However, I notice, in the supplementary figure 11G, that the (sedimentary accumulation rate) SAR of the studied section was mostly within the range of 2-4 cm/kyr before onset of Antarctica glaciation but at least doubled for many intervals after the glaciations. In a low SAR regime, factors like bioturbations tend to destroy shorter orbital signals and leave longer 100- and 400- bands signals. When SAR increased after glaciations at 33.7Ma, the bioturbations will have less "smoothing" effect on shorter orbital signals, which can potentially explain their appearance after the glaciations. Furthermore, because climate drying generally puts stress on biosphere, I tend to think that bioturbations would be less intense (less smoothing) in terrestrial sediments in a drier climate, as is recorded by this manuscript after onset of the glaciations. Therefore, both high SAR and drier climate/environments after glaciations would tend to promote preservation of short orbital cycles after ~33.7 Ma. So the observed orbital band changes from the studied sites can be a result of SAR and bioturbation variations, instead of the inferred climate reorganization! Note that SAR also increased right at 33.7 Ma for the ODP site 1218, which also recorded similar orbital band variations (Fig. 4F). So the orbital band variations as are recorded by the marine cores could also suffer from the same smoothing effect.

I feel that this alternative interpretation needs to be discussed and excluded in the revised version before one can make a judgement. Some convincing evidence supporting their argument would come from an East Asian terrestrial site which did not experience dramatic SAR variations across 33.7 Ma. If the reported orbital band variations can be also observed from such a record, it'd lend support for their interpretation.

Others:

I feel that tuning 100-kyr signals for the interval which have weaker variability can result in over tuning; so I suggest that the authors not tune those portion to short eccentricity.

Reviewer #3:

Remarks to the Author:

I read the revised manuscript and the authors' responses to comments of first circular reviews, although the authors tried to resolve the problems raised in previous reviews, I do not think they have properly addressed the comments #1, #2, #3, #4 of the first reviewer, either did not address comments #2, #3, #5 of the third reviewer at all. In addition, there are other issues I consider fundamental to the conclusions of this paper, that lead me to conclude that this paper is still premature for publication.

1. The authors did not describe the deposition sequence in detail, which casts doubt on their paleoclimate interpretation. In lines 114-117 they stated "The latest Eocene to earliest Oligocene succession studied here (383–570 m stratigraphic level within the Duitinggou section) is entirely within the Yehucheng Formation. The lithology comprises dark red mudstone, silty mudstone, and sandstone successions with distinct gypsiferous cyclic intercalations (Supplementary Fig. 3)." This description indicates there are sand deposits in the investigated sequence, therefore there were river channel deposits or sediments transported by a powerful process, which means that there were rapid deposit accumulations and/or hiatuses in this depositional sequence, however, there are not any descriptions of these potential hiatus in Figure 3 and Supplementary Figure 4. Even through the averaged sampling has a time resolution of 2-4 kyr, as the authors said, the hiatuses even on short

time scale will greatly change their interpretation. Thus, the authors should add detailed depositional analysis to their Tuiqinggou section.

Moreover, "The overlying Oligocene–Miocene Xianshuihe Formation consists mainly of light red mudstones that are intercalated with sandstone or conglomerate packages, while the underlying Early Eocene Xiliugou Formation consists of red massive sandstones (Supplementary Fig. 3)." (Lines 118–120). These sequences with coarser particle deposits such as sandstone or conglomerates suggest rapid accumulation and/or hiatus. These could cause uncertainty in the timescales, and thus a detailed analysis should be undertaken, unfortunately, lacking at present. In recent years, there are well known debates on timescales for the Cenozoic deposit sequences in the Qaidam Basin (to the west of the Lanzhou Basin) and the Ningxia Basin (to the east of the Lanzhou Basin) due to lack of detailed depositional environmental analysis and of independent age controls. I would consider that the Lanzhou Basin research needs to improve the quality of its timescales.

2. Circular reasoning. I am surprised that the authors used an astronomical tuning time series to do their spectral analysis, when the orbitally tuned timescale is artificial. Bringing orbital cycles into the time series before they do spectral analysis, therefore, is circular reasoning. I completely agree with the Reviewer 1 on this. Also, the independent time series (without orbitally tuning), which is indeed a suitable way to examine the orbital frequency, is relegated to the Supplementary Figure 12, not clearly unravel the frequency evolution during the EOT.

3. When we look at Figure 3, the magnetic susceptibility, the CIA and the STI do not present co-variation with humid and dry phase alternation; moreover, these records are conflicting at many times. This clearly indicates that these proxy indexes were not singularly determined by paleoclimate changes. I totally agree with comments of the first reviewer that "magnetic susceptibility is controlled by many factors, such as grain size, concentration, mineralogy, and its variations can be related to provenance, pedogenesis or chemical weathering, etc.". Unfortunately, the authors did not resolve these problems.

Since the aforementioned basic questions are not addressed in the revised manuscript, in addition to the questions not resolved in the previous round of review, I think conclusions in this paper are untenable. I do not think this research substantially improves our understanding on paleoclimate dynamics during the EOT on the northeastern Tibetan Plateau.

Reply to Reviewer #1

Comment #1

This manuscript has been clearly improved. However, I notice, in the supplementary figure 11G, that the (sedimentary accumulation rate) SAR of the studied section was mostly within the range of 2-4 cm/kyr before onset of Antarctica glaciation but at least doubled for many intervals after the glaciations. In a low SAR regime, factors like bioturbations tend to destroy shorter orbital signals and leave longer 100- and 400- bands signals. When SAR increased after glaciations at 33.7Ma, the bioturbations will have less “smoothing” effect on shorter orbital signals, which can potentially explain their appearance after the glaciations. Furthermore, because climate drying generally puts stress on biosphere, I tend to think that bioturbations would be less intense (less smoothing) in terrestrial sediments in a drier climate, as is recorded by this manuscript after onset of the glaciations. Therefore, both high SAR and drier climate/environments after glaciations would tend to promote preservation of short orbital cycles after ~33.7 Ma. So the observed orbital band changes from the studied sites can be a result of SAR and bioturbation variations, instead of the inferred climate reorganization! Note that SAR also increased right at 33.7 Ma for the ODP site 1218, which also recorded similar orbital band variations (Fig. 4F). So the orbital band variations as are recorded by the marine cores could also suffer from the same smoothing effect.

I feel that this alternative interpretation needs to be discussed and excluded in the revised version before one can make a judgement. Some convincing evidence supporting their argument would come from an East Asian terrestrial site which did not experience dramatic SAR variations across 33.7 Ma. If the reported orbital band variations can be also observed from such a record, it'd lend support for their interpretation.

Response: Thank you for pointing out to the potential impact of bioturbation on the observed orbital signature shift under lower sedimentation rates before ~33.7 Ma which, as correctly noted, broadly coincides with sedimentation rate changes in our new Lanzhou Basin record. This important aspect was underdeveloped in the previous text. In the newly revised manuscript, we devote substantial further text (lines 270-292 in the main text, and lines 211-261 in Supplementary Note 4) for this key issue. The included additional evidence suggests that bioturbation has little or no impact on the observed orbital signature shift. Based on a series of observations and comparisons, including

the sedimentology of Lanzhou palaeolake sediments, previous lake and marine bioturbation studies, and support from the East Asian Xining and Maoming basins, we conclude that the absence of precession and obliquity expression before 33.7 Ma in the Lanzhou χ record is unlikely to be caused by sedimentation rate and bioturbation variations. Consequently, the observed shift in orbital χ signature reflects an Asian climate reorganization in orbital forcing consistent with CO₂ decrease and major Antarctic glaciation across the global Oi-1 event (please see a full account in lines 211-261 in Supplementary Note 4).

Comment #2

I feel that tuning 100-kyr signals for the interval which have weaker variability can result in over tuning; so I suggest that the authors not tune those portion to short eccentricity.

Response: This is a good point, and it agrees with the actual method we used. In our tuning procedure, for a few intervals (e.g., ~32.7–33.6 Ma) with stronger 405-kyr variability than 100-kyr variability, the 405-kyr (rather than 100-kyr) matches were preferred with consideration of resulting sedimentation rate changes to avoid over-tuning during our refined tuning of the 100-kyr signal. Our tuning strategy therefore is fully consistent with this suggestion, which is now clarified in Supplementary Note 4 (lines 162-165).

Reply to Reviewer #3

Comment #1

I read the revised manuscript and the authors' responses to comments of first circular reviews, although the authors tried to resolve the problems raised in previous reviews, I do not think they have properly addressed the comments #1, #2, #3, #4 of the first reviewer, either did not address comments #2, #3, #5 of the third reviewer at all. In addition, there are other issues I consider fundamental to the conclusions of this paper, that lead me to conclude that this paper is still premature for publication.

Response: All comments raised by the three reviewers on the original submission were addressed extensively in our previous revision and were explained in detail our changes in the

response letter, including comments 1, 2, 3, and 4 of reviewer #1, and comments 2, 3, and 5 of reviewer #3 (see our previous revision and response letter for details). Moreover, three “other new” comments that reviewer #3 considered fundamental to the conclusions of our work, including circular reasoning in our orbital tuning strategy (comment 2), proxy validity (comment 3), sedimentary hiatuses (comment 1), and time scale uncertainties (comment 1), are reiterated comments raised by reviewers #1 and #2 on the original submission during the first round of review (comments 1–4 of reviewer #1, and minor comment of reviewer #2). We had addressed these (almost) completely in our previous revision, and reviewers #1 and #2 were satisfied with our revisions pending only two new (minor to modest) items. So, we have focused on the re-iteration of these comments by reviewer #3 as a signal that we had still not been clear enough in our responses, and that we needed to be even more explicit. We have therefore sought to further address them in detail, as outlined in the following:

Comment #2

The authors did not describe the deposition sequence in detail, which casts doubt on their paleoclimate interpretation. In lines 114-117 they stated “The latest Eocene to earliest Oligocene succession studied here (383–570 m stratigraphic level within the Duitinggou section) is entirely within the Yehucheng Formation. The lithology comprises dark red mudstone, silty mudstone, and sandstone successions with distinct gypsiferous cyclic intercalations (Supplementary Fig. 3).” This description indicates there are sand deposits in the investigated sequence, therefore there were river channel deposits or sediments transported by a powerful process, which means that there were rapid deposit accumulations and/or hiatuses in this depositional sequence, however, there are not any descriptions of these potential hiatus in Figure 3 and Supplementary Figure 4. Even through the averaged sampling has a time resolution of 2-4 kyr, as the authors said, the hiatuses even on short time scale will greatly change their interpretation. Thus, the authors should add detailed depositional analysis to their Tuiqinggou section.

Moreover, “The overlying Oligocene–Miocene Xianshuihe Formation consists mainly of light red mudstones that are intercalated with sandstone or conglomerate packages, while the underlying Early Eocene Xiliugou Formation consists of red massive sandstones (Supplementary Fig. 3).” (Lines 118-120). These sequences with coarser particle deposits such as sandstone or conglomerates suggest

rapid accumulation and/or hiatus. These could cause uncertainty in the timescales, and thus a detailed analysis should be undertaken, unfortunately, lacking at present. In recent years, there are well known debates on timescales for the Cenozoic deposit sequences in the Qaidam Basin (to the west of the Lanzhou Basin) and the Ningxia Basin (to the east of the Lanzhou Basin) due to lack of detailed depositional environmental analysis and of independent age controls. I would consider that the Lanzhou Basin research needs to improve the quality of its timescales.

Response: In our previous revision, we had already added significant new text (533 words, lines 99-138) to describe the geological background in detail. Doing so, we followed the same strategy and description used for the Xining Basin, as published in *Nature* by our corresponding author Dupont-Nivet et al. (2007). To create the requested further clarity, we have now even further expanded the text (lines 125-143 in the main text).

Our detailed outcrop observations in the field did not reveal any observable hiatuses in the late Eocene–early Oligocene Duitinggou section. Clearly, we cannot add unidentified hiatuses into Figure 3 and Supplementary Figure 4. The comment on sedimentary hiatuses, including “*Even through the averaged sampling has a time resolution of 2-4 kyr, as the authors said, the hiatuses even on short time scale will greatly change their interpretation*”, had been thoroughly addressed already. In the previous revision, we stated in the main text (lines 251-255 of the previously revised version and lines 261-269 in this further revision) and in the Supplementary Note 4 that potential short-duration hiatuses may cause one or several precession cycles to be (partially) missed in a few intervals, but are unlikely to have caused all obliquity and precession cycles to have been missed throughout the ~2-Myr interval between 35.6 and 33.6 Ma. Even though a few potential short-duration hiatuses may exist, they do not interfere with the clear orbital variations observed in both the Lanzhou and Xining χ records.

Including a discussion on potential hiatuses and their impacts on orbital variability was suggested originally by reviewers #1 and #2 (not reviewer #3). We have given considerable attention to it in our revision (see our responses to comment #3 of reviewer #1 and comment #5 of reviewer #2), and both reviewers #1 and #3 were satisfied by our previous revisions and responses.

The magnetostratigraphy for the overlying Oligocene–Miocene Xianshuihe Formation and associated uncertainties are discussed in detail in our published studies (Zhang et al., 2016, 2018), so here we do not repeat their established correlations with the GPTS. We already clarified extensively

on correlations of our new results to the GPTS with its associated uncertainties in our previous revision (see response to comment #1 of reviewer #1 in our previous revision), with which reviewer #1 is now satisfied. Moreover, our magnetostratigraphy is supported not only by previous and recent Lanzhou Basin magnetostratigraphic records, but also by magnetostratigraphic records from the neighboring Xining Basin (Fig. 2, Supplementary Fig. 4), which further adds to the robustness of the age model, as was agreed by reviewers #1 and #2.

Dupont-Nivet G., et al., 2007. Tibetan Plateau aridification linked to global cooling at the Eocene–Oligocene transition. *Nature* 445, 635–638, doi:10.1038/Nature05516.

Zhang, P., Ao, H., Dekkers, M.J., Li, Y.X., An, Z.S., 2016. Late Oligocene–Early Miocene magnetostratigraphy of the mammalian faunas in the Lanzhou Basin—environmental changes in the NE margin of the Tibetan Plateau. *Sci. Rep.* 6, 38023, doi: 38010.31038/srep38023.

Zhang, P., Ao, H., Dekkers, M.J., An, Z.S., Wang, L.J., Li, Y.X., Liu, S.H., Qiang, X.K., Chang, H., Zhao, H., 2018. Magnetostratigraphy of the Oligocene mammalian faunas in the Lanzhou Basin, Northwest China. *J. Asian Earth Sci.* 159, 24–33.

Comment #3

Circular reasoning. I am surprised that the authors used an astronomical tuning time series to do their spectral analysis, when the orbitally tuned timescale is artificial. Bringing orbital cycles into the time series before they do spectral analysis, therefore, is circular reasoning. I completely agree with the Reviewer 1 on this. Also, the independent time series (without orbital tuning), which is indeed a suitable way to examine the orbital frequency, is relegated to the Supplementary Figure 12, not clearly unravel the frequency evolution during the EOT.

Response: Magneto- and astro-chronology are combined routinely to develop refined timescales for marine records (for example, Coxall and Wilson, 2011; Drury et al., 2016; Holbourn et al., 2005; Lisiecki and Raymo, 2005; Pälike et al., 2001, 2006) and terrestrial records (for example, Prokopenko et al., 2006; Sun et al., 2006; An et al., 2011). In particular, astronomical tuning experts such as Hilgen, Hinnov, and Pälike have laid out the logic step-by-step in their papers to walk readers through the reasoning to demonstrate that it is not circular. When tuning or correlating, one works iteratively on increasingly shorter time scales, starting with the longest time frame as we did. Often these are biozones or magnetostratigraphic zones. Then, within the bio-magneto-constraints a suitable proxy record (here χ) is filtered on the long eccentricity period, which is the most stable orbital pacing throughout the Eocene–Oligocene (Pälike et al., 2001, 2006). As a check, filtering is also done on the

magnetostratigraphy or without time constraints, in the depth domain. If similar spectral structure arises, the filtering is robust and independent of the age constraints that may have gone into it. As a final step adopted here, filtering and tuning was done for short eccentricity cycles. Thus, the variable expression of obliquity and precession along our record do not stem from the tuning procedure because we did not tune for them (even though the resolution would allow it: 2–4 kyr is ~7 samples per precession cycle, which is easily sufficient for meaningful detection by spectral means). Accordingly, the appearance of obliquity and precession signals after ~33.7 Ma in the Lanzhou Basin is unlikely to be an artefact of the tuning. More importantly, the orbital χ shift at ~33.7 Ma is observed not only on the astronomical chronology (Fig. 4), but also consistently in the depth domain (Supplementary Fig. 11) and on the magnetostratigraphy (Supplementary Fig. 12). In all three records, the orbital expression is similar. Of course, in a tuned record the expression is more distinct; this is the whole point of tuning. In addition, a similar shift is also evident in the Xining χ record (Supplementary Fig. 14) and in the ODP Site 1218 benthic $\delta^{18}\text{O}$ record (Fig. 4) as suggested by our spectral analyses, and in the Maoming lithological record (South China) as suggested by cycle counting (Li et al., 2016), which lends support for the robustness of our observed orbital climate changes across the Oi-1 event.

The comment concerning circular reasoning was originally raised by reviewer #1. As suggested by reviewer #1, in our previous revision we conducted spectral analyses of the Lanzhou χ record on the untuned magnetostratigraphy (Supplementary Fig. 12) to document that the orbital shift is a robust feature and not a tuning-related artifact. Reviewer #1 was satisfied with our response. In this further revised version in responses to reviewer #3, this conclusion is further strengthened by added spectral analysis of the Lanzhou χ record in the depth domain (Supplementary Fig. 11), and by the similar orbital shift observed in other terrestrial and marine records. Also, Supplementary Note 4 addresses the circular reasoning issue in full.

- An, Z.S., Clemens, S.C., Shen, J., Qiang, X.K., Jin, Z.D., Sun, Y.B., Prell, W.L., Luo, J.J., Wang, S.M., Xu, H., Cai, Y.J., Zhou, W.J., Liu, X.D., Liu, W.G., Shi, Z.G., Yan, L.B., Xiao, X.Y., Chang, H., Wu, F., Ai, L., Lu, F.Y., 2011. Glacial-interglacial Indian summer monsoon dynamics. *Science* 333, 719-723.
- Coxall, H.K., Wilson, P.A., 2011. Early Oligocene glaciation and productivity in the eastern equatorial Pacific: Insights into global carbon cycling. *Paleoceanography* 26, PA2221, doi:2210.1029/2010PA002021.
- Drury, A.J., John, C.M., Shevenell, A.E., 2016. Evaluating climatic response to external radiative forcing during the late Miocene to early Pliocene: new perspectives from eastern equatorial Pacific (IODP U1338) and North Atlantic

- (ODP 982) locations. *Paleoceanography* 31, 167-184.
- Holbourn, A., Kuhnt, W., Schulz, M., Erlenkeuser, H., 2005. Impacts of orbital forcing and atmospheric carbon dioxide on Miocene ice-sheet expansion. *Nature* 438, 483-487.
- Lisiecki, L.E., Raymo, M.E., 2005. A Pliocene-Pleistocene stack of 57 globally distributed benthic $\delta^{18}\text{O}$ records. *Paleoceanography* 20, PA1003, doi:10.1029/2004PA001071.
- Li, Y.X., Jiao, W.J., Liu, Z.H., Jin, J.H., Wang, D.H., He, Y.X., Quan, C., 2016. Terrestrial responses of low-latitude Asia to the Eocene-Oligocene climate transition revealed by integrated chronostratigraphy. *Clim Past* 12, 255-272.
- Pälike, H., Norris, R.D., Herrle, J.O., Wilson, P.A., Coxall, H.K., Lear, C.H., Shackleton, N.J., Tripathi, A.K., Wade, B.S., 2006. The heartbeat of the Oligocene climate system. *Science* 314, 1894-1898.
- Pälike, H., Shackleton, N.J., Rohl, U., 2001. Astronomical forcing in Late Eocene marine sediments. *Earth Planet. Sci. Lett.* 193, 589-602.
- Prokopenko, A.A., Hinnov, L.A., Williams, D.F., Kuzmin, M.I., 2006. Orbital forcing of continental climate during the Pleistocene: a complete astronomically tuned climatic record from Lake Baikal, SE Siberia. *Quat. Sci. Rev.* 25, 3431-3457.
- Raymo, M.E., Lisiecki, L.E., Nisancioglu, K.H., 2006. Plio-pleistocene ice volume, Antarctic climate, and the global $\delta^{18}\text{O}$ record. *Science* 313, 492-495.
- Sun, Y.B., Clemens, S.C., An, Z.S., Yu, Z.W., 2006. Astronomical timescale and palaeoclimatic implication of stacked 3.6-Myr monsoon records from the Chinese Loess Plateau. *Quat. Sci. Rev.* 25, 33-48.

Comment #4

When we look at Figure 3, the magnetic susceptibility, the CIA and the STI do not present co-variation with humid and dry phase alternation; moreover, these records are conflicting at many times. This clearly indicates that these proxy indexes were not singularly determined by paleoclimate changes. I totally agree with comments of the first reviewer that “magnetic susceptibility is controlled by many factors, such as grain size, concentration, mineralogy, and its variations can be related to provenance, pedogenesis or chemical weathering, etc.” Unfortunately, the authors did not resolve these problems.

Response: Our χ record has a much higher resolution than the CIA and the STI records. The high-resolution χ record contains evidence of both long-term and orbital-scale variability, whereas the low-resolution CIA and the STI records can only be used to make inferences on long-term variability (see lines 215-218 and 311-314 in the main text of this revision for detailed clarifications). We, as well as reviewer #1, noted this. Reviewer #1 then suggested that palaeoclimate records with too large a difference in resolution should not be used for detailed comparison purposes. Reviewer #1 commented on the original submission: “229-230: *the Xining paleotemperature record has too low resolution to compare with your data here*”. We agreed and modified this accordingly in the previous revision. Therefore, as suggested by reviewer #1, it is not unusual that the significantly

lower-resolution CIA, K_2O/Al_2O_3 , and STI records cannot be compared with the χ record over orbital (precessional) timescales for a few intervals (see lines 215-218 and 311-314 in the main text of revision 2 for our detailed clarification of this issue). In our previous revision, we devoted a large amount of text (lines 163-186 in the main text and Supplementary Figs. 7–10) to address this issue, as guided by reviewer #1 (see our detailed responses to comment 4 of reviewer #1 in our response letter). Following the suggestions of reviewer #1, our major interpretations were based on the high-resolution χ record that fully samples on precessional time scales, and we use the lower-resolution chemical weathering proxies only to support observed long-term trend changes in the high-resolution χ record.

Given that reviewer #1 was satisfied with our response, we cannot really follow reviewer #3's comment that "*Unfortunately, the authors did not resolve these problems*". Regardless, we make further efforts to better clarify the justification of χ for hydroclimate variability in the present revision, in a completely new section in Supplementary Note 3.

In summary, the three "other new" comments of reviewer #3 on our previous revision all derived from points originally raised by reviewers #1 and/or #2 on the original submission. We addressed these comments extensively in our previous revision. Reviewer #1 was satisfied by our revisions with only two minor to modest comments remaining, while reviewer #2 was fully satisfied with our revisions. Reviewer #3 then rekindled some of the discussion that were laid to rest, which likely indicates that we had not been entirely clear enough at all levels. Hence, we have made further efforts to address the issues with even more clarity in the present revision. None of them changes our results and conclusions.

Reviewers' Comments:

Reviewer #1:

Remarks to the Author:

I am generally satisfied with the revised manuscript, which has been much clearer and more convincing after addressing the previous comments. I do want to make one more point, though. Lack or weak precession cycles in sedimentary records can be caused by environmental smoothing, which consists of many factors including bioturbation (Su et al., 2019G3 named a few key factors). So I suggest that the authors change bioturbation to environmental smoothing.

Ironically, the best evidence supporting negligible environmental smoothing effect is to detect clear precession cycles prior glaciation using different environmental proxies because different proxies might have different sensitivities to insolation forcing. This might take many tests and the environment proxy has to be the right one, so this is definitely beyond the scope of this manuscript. But luckily, Su et al. did show a case for the Pliocene interval. So the approach in Su et al. may be applied in the future. I suggest that the authors state, instead of only increasing sampling resolution (as is suggested in line269), trying more environmental proxies would also be necessary in the future. It might also be a good idea, and adds more confidence, to mention that similar orbital band changes in NE Tibetan Plateau had happened more than once before. Nie et al. (2017) documented a similar orbital-band variation intensification in precipitation proxies at ca. 8.5 Ma, which was tied to possible Antarctica cooling and Tibetan uplift forcing (the two could be genetically linked): eccentricity, obliquity and precession all popped up after 8.5 Ma but weak before. Nie et al. (2017) didn't tune their records to eccentricity; if they do so, similar pattern to this ms would appear: clear eccentricity cycles before cooling and all orbital cycles after cooling at 8.5 Ma. So it seems a robust feature that NE Tibetan plateau environmental variations at orbital bands are strong during colder climate. In summary, I feel that this version is much stronger now and I feel that intensified orbital band variations in terrestrial environment would be of interests to a broad audience.

Reviewer #3:

Remarks to the Author:

Ao et al. have tried their very best to address the questions raised in previous rounds of reviews. I appreciate that some of the problems have been resolved, and the quality of this paper has been improved. In my previous reviews, I wavered over the question of whether minor or major revisions to this paper would be necessary, but now after reading the revised paper, the supplementary information and the authors' responses to previous comments, I am sad to say that the essential scientific evidence for this paper is still not well presented. Some of the necessary information remained concealed - unintentionally, I hope.

Therefore, I am not convinced that the authors have properly addressed the questions in the previous reviewer comments, including both the first and second rounds of comments.

I will not add further comments. The authors can refer the comments in the previous two reviews. My main concerns are:

1. There are sand layers in Yehucheng formation in Lanzhou Basin in the authors' previous description (Zhang Peng, 2015) and others' investigations (Gu Yansheng et al., 2000; Yue Leping et al., 2003; Li Zhichao et al., 2016; Wu Fuli et al., 2018; and the references thereafter), but the sand layers have totally disappeared in the revised paper (line 123-150; Figure 2, Yongdeng Section and Dutinggou section). We know that such sandy sediments/layers are vital in building an accurate and reliable time scale and in the paleoclimate interpretation in the deposition sequences. However, this important information was ignored. I feel the "high-resolution" time scale should be further assessed (line 196-

198, an average resolution of 2.5 kyr between 31.0 and 35.6 Ma, with a typical resolution of 2-4 kyr before 33.65 Ma and 1-3.5 kyr after 33.65 Ma).

2. The paleoclimate implications of CIA, K_2O/Al_2O_3 and STI are sediment source based and are significantly influenced by grain size distribution: the bulk samples analyzed in this study brought multiple interpretations in the paleoclimate records. These questions are still not well addressed in the revised manuscript. Similar to the geochemical proxies, the authors explain the magnetic susceptibility as follow: "In addition, previous studies have shown that wetter climate generally enhances regional vegetation cover, which stabilizes the landscape. This is anticipated to have decreased detrital magnetic mineral input 79,80 (Supplementary Fig. 10). In contrast, arid climates can be associated with increased detrital catchment magnetic mineral input (Supplementary Fig. 10)." (Supplementary Information for Ao et al., Line 114-117). This interpretation is too speculative and vague. Indeed, a wetter climate can also enhance the movement of detrital magnetic minerals into lake sediments (Thompson and Oldfield, 1986. Chapter 8, 9, 10).

In Supplementary Figure 8, relationships between the magnetic properties are apparently circular reasoning, since these are proxy indicators based fundamentally on magnetic minerals, so the relationship between them is not helpful in explaining the paleoclimatic implications of the geochemical and environment magnetic proxies employed in this study. I would like to see the statistical relationships between magnetic susceptibility, CIA, K_2O/Al_2O_3 and STI, which is of paramount importance for scrutinizing the paleoclimatic implications of these proxy indices, i.e. whether they were climatic driven, co-varying and statistically significant. Unfortunately, these discussions were not present in the main text nor in the Supplementary Information. I feel these proxies have been over-interpreted here. The proxy time series shifted at ~ 33.7 Ma could be caused by earth surface processes such as rerouting of the infilling rivers or migration of the shallow lake, which can commonly occur in semiarid and arid climates.

3. The untuned time series is extremely important for investigating the Milankovitch/astronomical cycles from geological records. For the untuned timescale, I would like very much to see the results from the 2n-Multi-Taper Method (MTM) used in this study, but the authors have presented results of the wavelet transfer in the main text and the Supplementary Information only. The results of wavelet transfer seem to me to be ambiguous. I do not understand why the authors did not present results of the MTM spectral analysis on the untuned time series, even if only in the Supplementary Information. The results from the tuned time series are based on 47 age control points (Supplementary Table 1. Age correlation points were used to derive the adopted astronomically tuned age model.), but in the previous version, the authors used 45 age control points, so why were two more added? I agree with Reviewer #1 that using too many age control points has brought artifact cyclicity into the results, although the authors did not agree with this point. In order to identify the statistically significant spectral peaks in the untuned time series, noise estimation must be undertaken. Did the spectral peaks rise above the confidence curve in the untuned time series? Were the untuned spectra considered statistically significant? Unfortunately, this important information is not presented.

4. I feel that the lines 323-362 in the main text have too much discussion on the forcing mechanisms of arid-humid evolution in Lanzhou region. Most of these conclusions are already well-known (Please see Li J. X., et al., 2018, Nature Communications, and the references thereafter). I suggest shortening this paragraph.

5. The first sentence of abstract wrote "the EOT-1 cooling event at 34.1–39.9 Ma and the Oi-1 glaciation" (line 43). I believe the "39.9 Ma" is a spelling error, but I am surprised that so many authors did not notice the mistake in this crucial number. There are also several references cited that are not pertinent to this paper.

Reply to Reviewer #1

Comment #1

I am generally satisfied with the revised manuscript, which has been much clearer and more convincing after addressing the previous comments. I do want to make one more point, though. Lack or weak precession cycles in sedimentary records can be caused by environmental smoothing, which consists of many factors including bioturbation (Su et al., 2019G3 named a few key factors). So I suggest that the authors change bioturbation to environmental smoothing.

Ironically, the best evidence supporting negligible environmental smoothing effect is to detect clear precession cycles prior glaciation using different environmental proxies because different proxies might have different sensitivities to insolation forcing. This might take many tests and the environment proxy has to be the right one, so this is definitely beyond the scope of this manuscript. But luckily, Su et al. did show a case for the Pliocene interval. So the approach in Su et al. may be applied in the future. I suggest that the authors state, instead of only increasing sampling resolution (as is suggested in line269), trying more environmental proxies would also be necessary in the future.

Response: *Thanks for your previous insightful comments that helped to improve our manuscript. We have changed the wording 'bioturbation' to 'environmental smoothing', and stated the importance of adding multiple environmental proxies in future studies (lines 284, 285, and 302).*

Comment #2

It might also be a good idea, and adds more confidence, to mention that similar orbital band changes in NE Tibetan Plateau had happened more than once before. Nie et al. (2017) documented a similar orbital-band variation intensification in precipitation proxies at ca. 8.5 Ma, which was tied to possible Antarctica cooling and Tibetan uplift forcing (the two could be genetically linked): eccentricity, obliquity and precession all popped up after 8.5 Ma but weak before. Nie et al. (2017) didn't tune their records to eccentricity; if they do so, similar pattern to this ms would appear: clear eccentricity cycles before cooling and all orbital cycles after cooling at 8.5 Ma. So it seems a robust feature that NE Tibetan plateau environmental variations at orbital bands are strong during colder climate.

In summary, I feel that this version is much stronger now and I feel that intensified orbital band variations in terrestrial environment would be of interests to a broad audience.

Response: *We have now mentioned the similar orbital shift observed in the Qaidam Basin at ~8.5 Ma when global climate cooled and Antarctic glaciation intensified in the main text to add more confidence to our findings (lines 309-311).*

Reply to Reviewer #3

Comment #1

There are sand layers in Yehucheng formation in Lanzhou Basin in the authors' previous description (Zhang Peng, 2015) and others' investigations (Gu Yansheng et al., 2000; Yue Leping et al., 2003; Li Zhichao et al., 2016; Wu Fuli et al., 2018; and the references thereafter), but the sand layers have totally disappeared in the revised paper (line 123-150; Figure 2, Yongdeng Section and Dutinggou section). We know that such sandy sediments/layers are vital in building an accurate and reliable time scale and in the paleoclimate interpretation in the deposition sequences. However, this

important information was ignored. I feel the “high-resolution” time scale should be further assessed (line 196-198, an average resolution of 2.5 kyr between 31.0 and 35.6 Ma, with a typical resolution of 2-4 kyr before 33.65 Ma and 1-3.5 kyr after 33.65 Ma).

Response: *The Yehucheng Formation in the studied Duitinggou section is dominated by mudstone, siltstone, and gypsiferous beds, which is supported by the mean grain size record (Fig. 2). There are only four 1–3 m thick grayish white fine sand beds in the ~190 m thick Yehucheng Formation studied here, which is also supported by the mean grain size record (Fig. 2), although more sand beds are present above and below the presently-studied interval. The four grayish white fine sand beds were deposited under subaqueous alluvial to shallow playa lake conditions, which facilitated generally continuous deposition although with a slightly coarser grain size and potential small hiatuses. Thus, these sand beds are unlikely to influence substantially the time scale framework (including accuracy and resolution) and the palaeoclimate reconstruction, both of which are established principally from the dominant mudstone, siltstone, and gypsiferous beds. The requested detailed information on the sand beds is now included in the revised manuscript (lines 146-149). Furthermore, we also now describe the depositional sequence in more detail (lines 122-161), and have added a new mean grain size record for the section (Figs. 2 and 3), which further strengthens our palaeoclimate interpretations.*

Comment #2

The paleoclimate implications of CIA, K_2O/Al_2O_3 and STI are sediment source based and are significantly influenced by grain size distribution: the bulk samples analyzed in this study brought multiple interpretations in the paleoclimate records. These questions are still not well addressed in the revised manuscript. Similar to the geochemical proxies, the authors explain the magnetic susceptibility as follow: “In addition, previous studies have shown that wetter climate generally enhances regional vegetation cover, which stabilizes the landscape. This is anticipated to have decreased detrital magnetic mineral input 79,80 (Supplementary Fig. 10). In contrast, arid climates can be associated with increased detrital catchment magnetic mineral input (Supplementary Fig. 10).” (Supplementary Information for Ao et al., Line 114-117). This interpretation is too speculative and vague. Indeed, a wetter climate can also enhance the movement of detrital magnetic minerals into lake sediments (Thompson and Oldfield, 1986. Chapter 8, 9, 10).

In Supplementary Figure 8, relationships between the magnetic properties are apparently circular reasoning, since these are proxy indicators based fundamentally on magnetic minerals, so the relationship between them is not helpful in explaining the paleoclimatic implications of the geochemical and environment magnetic proxies employed in this study. I would like to see the statistical relationships between magnetic susceptibility, CIA, K_2O/Al_2O_3 and STI, which is of paramount importance for scrutinizing the paleoclimatic implications of these proxy indices, i.e. whether they were climatic driven, co-varying and statistically significant. Unfortunately, these discussions were not present in the main text nor in the Supplementary Information. I feel these proxies have been over-interpreted here. The proxy time series shifted at ~33.7 Ma could be caused by earth surface processes such as rerouting of the infilling rivers or migration of the shallow lake, which can commonly occur in semiarid and arid climates.

Response: *First, CIA and K_2O/Al_2O_3 vary consistently during the late Eocene–early Oligocene (Fig. 3F–G), and correlate linearly with each other (Supplementary Fig. 14). Second, they correlate poorly with mean grain size (Supplementary Fig. 14). Third, grain size does not shift across Oi-1*

(Fig. 3B). Thus, the CIA and K_2O/Al_2O_3 shifts across Oi-1 are unlikely to have been caused by changes in the grain-size distribution, but suggest a decrease in chemical weathering intensity with lower temperature and precipitation.

We have now removed the discussion of potential links between detrital magnetic mineral inputs and climate. The magnetic susceptibility changes can be still explained reasonably well by the discussion retained in the manuscript. Gypsum dilutes the magnetic susceptibility expression because it is diamagnetic; detrital magnetic minerals are also partially dissolved post-depositionally in anoxic depositional environments with higher precipitation. In oxic environments with ephemeral subaerial exposure, magnetic minerals are preserved and could even be formed pedogenically (see Supplementary Note 3 for details; Supplementary Fig. 10). Thus, susceptibility increases across Oi-1, which is consistent with the depicted scenario.

We have added details of the statistical relationship between magnetic susceptibility, CIA, and K_2O/Al_2O_3 as suggested (Supplementary Fig. 14). Their good correlations with each other and poor correlation with mean grain size record support our palaeoclimatic interpretations. The absence of a grain size shift, prevailing occurrence of χ , SIRM, HIRM, CIA, and K_2O/Al_2O_3 shifts across Oi-1, and well-known forcing dynamics suggest that their changes at ~ 33.7 Ma are reasonable indicators of climate changes. We have removed STI from the proxy records, because it has a higher correlation ($R = 0.58$) with grain size than CIA and K_2O/Al_2O_3 ($R = 0.16$), and a lower correlation ($R = 0.53$) with K_2O/Al_2O_3 than CIA ($R = 0.79$) (Fig. 1 below and Supplementary Figure 14). However, the removal does not influence our chemical weathering and palaeoclimate interpretations, which are strongly based on magnetic susceptibility, HIRM, SIRM, CIA, and K_2O/Al_2O_3 records.

Figure 1. Relationship of mean grain size versus STI, and STI versus CIA for the Duitinggou section.

Comment #3

The untuned time series is extremely important for investigating the Milankovitch/astronomical cycles from geological records. For the untuned timescale, I would like very much to see the results

from the 2π -Multi-Taper Method (MTM) used in this study, but the authors have presented results of the wavelet transfer in the main text and the Supplementary Information only. The results of wavelet transfer seem to me to be ambiguous. I do not understand why the authors did not present results of the MTM spectral analysis on the untuned time series, even if only in the Supplementary Information. The results from the tuned time series are based on 47 age control points (Supplementary Table 1. Age correlation points were used to derive the adopted astronomically tuned age model.), but in the previous version, the authors used 45 age control points, so why were two more added? I agree with Reviewer #1 that using too many age control points has brought artifact cyclicity into the results, although the authors did not agree with this point. In order to identify the statistically significant spectral peaks in the untuned time series, noise estimation must be undertaken. Did the spectral peaks rise above the confidence curve in the untuned time series? Were the untuned spectra considered statistically significant? Unfortunately, this important information is not presented.

Response: *We now include a 2π -Multi-taper method (MTM) power spectrum for the Lanzhou χ record on the untuned magnetochronology, in addition to the spectral evolutionary spectrum (see Supplementary Fig. 12). Precession and obliquity cycles appear after 33.7 Ma in the χ record on the untuned magnetochronology, consistent with the observed feature on the combined magneto- and astro-chronology (Fig. 4). Yes, we used 45 age control points in the initial version and 47 in the following revised versions to establish the astronomical chronology. Based on related suggestions from reviewer #1, we adjusted the astronomical time series to improve simultaneously the consistency of both the 405-kyr and 100-kyr χ components with their target curves, with palaeomagnetic reversal ages, and with sedimentation rate variability. This slightly adjusted astronomical chronology involved 47 age control points, two more than the initially used. However, the resulting orbital features are identical to those initially observed.*

The observed appearance of obliquity and precession signals after ~ 33.7 Ma in the Lanzhou χ record is unlikely to be an artefact of the tuning. It is observed not only on the combined magneto- and astro-chronology (Fig. 4), but also consistently in the depth domain (Supplementary Fig. 11) and on the magnetochronology (Supplementary Fig. 12). In all three records, the orbital expression is similar. Of course, orbital expression is more distinct in a tuned record and non-orbital signals are weakened, which is the whole point of tuning. Furthermore, a similar shift is also evident in the Xining χ record (Supplementary Fig. 15) and in the ODP Site 1218 benthic $\delta^{18}\text{O}$ record (Fig. 4) as suggested by recent wavelet spectrogram (De Vleeschouwer et al., 2017) and our spectral analyses, and in the Maoming lithological record (South China) as suggested by cycle counting (Li et al., 2016), which lends support to the robustness of our observed orbital climate changes across the Oi-1 event. In addition, a similar hydroclimate transition from eccentricity to combined eccentricity, obliquity, and precession cycles was also observed in the Qaidam Basin, NE Tibetan Plateau, at ~ 8.5 Ma when global climate cooled and Antarctic glaciation intensified (Nie et al., 2017), which provides additional support as noted by reviewer #1.

De Vleeschouwer, D., Vahlenkamp, M., Crucifix, M., Pälike, H., 2017. Alternating Southern and Northern Hemisphere climate response to astronomical forcing during the past 35 m.y. *Geology* 45, 375-378.

Li, Y.X., Jiao, W.J., Liu, Z.H., Jin, J.H., Wang, D.H., He, Y.X., Quan, C., 2016. Terrestrial responses of low-latitude Asia to the Eocene-Oligocene climate transition revealed by integrated chronostratigraphy. *Clim Past* 12, 255-272.

Nie, J.S., Garzzone, C., Su, Q.D., Liu, Q.S., Zhang, R., Heslop, D., Necula, C., Zhang, S.H., Song, Y.G. and Luo, Z., 2017. Dominant 100,000-year precipitation cyclicity in a late Miocene lake from northeast Tibet. *Science*

Comment #4

I feel that the lines 323-362 in the main text have too much discussion on the forcing mechanisms of arid-humid evolution in Lanzhou region. Most of these conclusions are already well-known (Please see Li J. X., et al., 2018, Nature Communications, and the references thereafter). I suggest shortening this paragraph.

Response: *Detailed discussion on the forcing mechanisms is essential for the readers to better understand the Asian hydroclimate changes across Oi-1. Although most of the forcing dynamics are well-known, how they work specifically across Oi-1 remain unexplored. Thus, we deem such detailed discussion useful. However, related references have been included, including Li et al. (2018) and other references.*

Comment #5

The first sentence of abstract wrote “the EOT-1 cooling event at 34.1–39.9 Ma and the Oi-1 glaciation” (line 43). I believe the “39.9 Ma” is a spelling error, but I am surprised that so many authors did not notice the mistake in this crucial number. There are also several references cited that are not pertinent to this paper.

Response: *Thanks for pointing out this typo. It has been corrected. We undertook a further check on the spelling and references and have removed some less relevant citations.*

Reviewers' Comments:

Reviewer #4:

Remarks to the Author:

Seeing this manuscript for the first time in this second round of the review process, I prefer to limit my comments on main issues and – as asked by the editor – to assess whether the Rev#3's concerns have been addressed sufficiently (Rev#3 is obviously not available for another review).

First of all, I find the story presented in this work fascinating. The authors present convincing geochemical and magnetic proxy results indicating a climate shift at the NE Tibetan Plateau region, probably to drier/colder conditions, approximately contemporaneous with the major Oi-1 Antarctic glaciation at 33.8-33.6 Ma. The well-determined polarity boundaries C13r-C13n and C13n-C12r in their new Lanzhou section are helpful here. According to the shift in the K2O/A2O3 ratio and the CIA it could be few 100-kyr later (which, however, is not explicitly discussed in the paper). The authors furthermore claim that time series analysis of magnetic susceptibility (χ) provides robust evidence for the appearance of obliquity and precession cycles immediately after the Oi-1, in addition to continuous eccentricity pacing. This interpretation might be correct, but I'm a bit irritated by the authors' handling of uncertainties, which they persistently tend to reject or not to address.

Age model and spectral results:

A high-resolution age model and a sufficiently linear time axis are indispensable to trace the orbital signatures and pin the transition to Oi-1 and obtain clear obliquity and precession signals. The astronomical tuning with magnetostratigraphic age markers is done by long and short eccentricity cycles. Uncertainties in fixing conflicts with polarity boundaries exceed 100 kyr which is a lot in regard of the interpretations based on the age model. It is mentioned that >50 tuning alternatives have been tested, but there is no information to which extent these alternative options differ. Most surprising to me are the very clean and continuous obliquity and precession signals in the WPS after tuning (and the existence in the untuned model and depth domain). The sampled sequence are deposits varying in facies between perennial lake, mudflat and alluvial fan. The astronomically tuned age model that averages on a 100-kyr scale shows a variation in sediment accumulation rates (SAR) by a factor of 3; magnetostratigraphy alone, averaging on longer times, shows nearly 2-fold differences in the SAR. It's therefore clear that on a shorter scale, at which obliquity and precession are resolved, the SAR will vary even more. With such variability, how can it be explained that obliquity and even precession show up clearly in a WPS result?

Rev#3 raised the uncertainty in the age model in his/her comment #3. The authors have addressed this concern by adding spectral results for the untuned age model. This is fine, but does bring in the necessary discussion on uncertainties.

Environmental meaning of magnetic proxy parameters:

I agree with Rev#3's concern that the environmental interpretation of the concentration-dependent magnetic parameters (χ , SIRM, HIRM) involves speculation. The authors changed their previous model from catchment-related weathering to dissolution in anoxic lake conditions, and also added a correlation analysis with the geochemical proxies. While χ correlates moderately with the CIA, the correlation with the K2O/A2O3 ratio is weak. I think the dissolution model is too simple. However, this does not mean that χ cannot be used for spectral analysis and tuning, and it also does not mean that χ has no significant climatic meaning. Besides possible dissolution, how about authigenic formation during mudflat conditions, even by more than a single process? Groundwater level change and formation of surface water during dry and wet mudflat conditions could lead to strong short-scale variations in magnetic mineral properties. Even catastrophic flood events should be considered. The facies variability is also of importance for the spectral results. I appreciate the additional presentation of lithological information in the revised manuscript (which is partly related to Rev#3's comment#1).

Despite the probably existing complexity in the χ -variation, let's buy the reasonable interpretation that high χ stands for more humid, and low χ for drier conditions. It is striking that the χ -variability is highest before the Oi-1 ranging from maximum to minimum values of the entire sequence, indicating that conditions varied from perennial lake to alluvial fan, likely including dry and wet mudflat states. This will cause a lot of noise in the χ -series, and in the view of spectral analysis the sampling resolution of few kyr is certainly strongly distorted by this variability. Did you ever consider to analyze the evolution of means and variances, and non-linear time series analysis such as empirical mode decomposition (e.g., by the Huang-Hilbert method). This could reveal features in the variability beyond the classical linear methods?

Final comment and recommendation:

My comments above should not be understood as a principle criticism against the authors' interpretations. Actually, I like the paper, but I'm perplex by the certainty and unambiguity in which it is presented. Remaining uncertainty is no reason for rejection, this would kill 99% of all papers in geoscience. My recommendation to the editor is to support the paper for publication, but to urge the authors to be more conservative and discuss/mention the uncertainties in a fair way, leaving open questions open for future work and discussion.

Erwin Appel, 2020-4-26

Reply to Reviewer #4

Comment #1

First of all, I find the story presented in this work fascinating. The authors present convincing geochemical and magnetic proxy results indicating a climate shift at the NE Tibetan Plateau region, probably to drier/colder conditions, approximately contemporaneous with the major Oi-1 Antarctic glaciation at 33.8-33.6 Ma. The well-determined polarity boundaries C13r-C13n and C13n-C12r in their new Lanzhou section are helpful here. According to the shift in the K₂O/A₂O₃ ratio and the CIA it could be few 100-kyr later (which, however, is not explicitly discussed in the paper). The authors furthermore claim that time series analysis of magnetic susceptibility (χ) provides robust evidence for the appearance of obliquity and precession cycles immediately after the Oi-1, in addition to continuous eccentricity pacing. This interpretation might be correct, but I'm a bit irritated by the authors' handling of uncertainties, which they persistently tend to reject or not to address.

Response: We are grateful for the positive view of the reviewer and helpful suggestions, including the uncertainty issue. We now include significant additional text in which we discuss age uncertainties in detail (Supplementary Note 4, lines 276–320; red vertical lines in Supplementary Fig. 16F; Supplementary Table 1; see responses to Comment #2). The reviewer correctly notes the presence of 100-kyr cycles in the CIA and K₂O/Al₂O₃ records, which we did not discuss earlier. We now mention this in the main text (lines 244–246).

Comment #2

Age model and spectral results:

A high-resolution age model and a sufficiently linear time axis are indispensable to trace the orbital signatures and pin the transition to Oi-1 and obtain clear obliquity and precession signals. The astronomical tuning with magnetostratigraphic age markers is done by long and short eccentricity cycles. Uncertainties in fixing conflicts with polarity boundaries exceed 100 kyr which is a lot in regard of the interpretations based on the age model. It is mentioned that >50 tuning alternatives have been tested, but there is no information to which extent these alternative options differ. Most surprising to me are the very clean and continuous obliquity and precession signals in the WPS after tuning (and the existence in the untuned model and depth domain). The sampled sequence are deposits varying in facies between perennial lake, mudflat and alluvial fan. The astronomically tuned age model that averages on a 100-kyr scale shows a variation in sediment accumulation rates (SAR) by a factor of 3; magnetostratigraphy alone, averaging on longer times, shows nearly 2-fold differences in the SAR. It's therefore clear that on a shorter scale, at which obliquity and precession are resolved, the SAR will vary even more. With such variability, how can it be explained that obliquity and even precession show up clearly in a WPS result? Rev#3 raised the uncertainty in the age model in his/her comment #3. The authors have addressed this concern by adding spectral results for the untuned age model. This is fine, but does bring in the necessary discussion on uncertainties.

Response: Thank you for this specific suggestion to help us to better handle age uncertainties, which are now discussed in much more detail in Supplementary Note 4 (lines 276–320 in the Supplementary Information). The depth and age uncertainties of tie points are also included in Supplementary Fig. 16F (red vertical lines) and Supplementary Table 1. As suggested, we also add related information on the considered > 50 tuning options (lines 276–320 in the Supplementary Information). We agree with reviewer #4 that the SAR will vary more significantly over obliquity and precession time scales, particularly if we further tune our eccentricity-based astronomical time scale to obliquity and precession cycles. This further tuning will intensify the obliquity and precession cycles in the Lanzhou

χ_{lf} record. However, to avoid ‘over-tuning’, we did not tune our eccentricity-based astronomical time scale to obliquity and precession cycles, particularly because these shorter cycles are absent in the Lanzhou χ_{lf} record in the original depth domain and in the untuned magnetochronology before 33.7 Ma (Supplementary Figs. 14–15). Not tuning to obliquity and precession resulted in increased age uncertainties for these shorter cycles and simultaneously smoothed the SAR variability. These factors likely weakened the obliquity and precession expression and resulted in non-orbital bands around obliquity and/or precession bands. However, they did not remove the original obliquity and precession signals, which are clear in the eccentricity-based astronomical time scale, the untuned magnetochronology, and the original depth domain (Fig. 4; Supplementary Figs. 14–15). Our minimum tuning strategy with eccentricity correlations decreased the age uncertainties and intensified the untuned obliquity and precession bands relative to the magnetochronology. This suggests that the appearance of obliquity and precession cycles after ~33.7 Ma is a robust feature, which is not removed by the smoothed SAR variability and accompanying age uncertainties.

Comment #3

Environmental meaning of magnetic proxy parameters:

I agree with Rev#3’s concern that the environmental interpretation of the concentration-dependent magnetic parameters (χ , SIRM, HIRM) involves speculation. The authors changed their previous model from catchment-related weathering to dissolution in anoxic lake conditions, and also added a correlation analysis with the geochemical proxies. While χ correlates moderately with the CIA, the correlation with the K_2O/A_2O_3 ratio is weak. I think the dissolution model is too simple. However, this does not mean that χ cannot be used for spectral analysis and tuning, and it also does not mean that χ has no significant climatic meaning. Besides possible dissolution, how about authigenic formation during mudflat conditions, even by more than a single process? Groundwater level change and formation of surface water during dry and wet mudflat conditions could lead to strong short-scale variations in magnetic mineral properties. Even catastrophic flood events should be considered. The facies variability is also of importance for the spectral results. I appreciate the additional presentation of lithological information in the revised manuscript (which is partly related to Rev#3’s comment#1). Despite the probably existing complexity in the χ -variation, let’s buy the reasonable interpretation that high χ stands for more humid, and low χ for drier conditions. It is striking that the χ -variability is highest before the Oi-1 ranging from maximum to minimum values of the entire sequence, indicating that conditions varied from perennial lake to alluvial fan, likely including dry and wet mudflat states. This will cause a lot of noise in the χ -series, and in the view of spectral analysis the sampling resolution of few kyr is certainly strongly distorted by this variability. Did you ever consider to analyze the evolution of means and variances, and non-linear time series analysis such as empirical mode decomposition (e.g., by the Huang-Hilbert method). This could reveal features in the variability beyond the classical linear methods?

Response: Thank you for this helpful comment. We have added FORC diagrams, low-temperature magnetic measurements, a frequency-dependent magnetic susceptibility record, and morphological and mineral analyses of magnetic extracts to better reveal the magnetic mineralogy and to interpret more precisely the likely mechanism of χ_{lf} variability for the Lanzhou Basin fluvial-lacustrine sediments (Supplementary Figs. 8–12; Supplementary Note 3). In addition to magnetic mineral dissolution, we now also include dilution by non-magnetic gypsum and limited subaerial pedogenic magnetic mineral formation to interpret the low χ_{lf} in the greyish white gypsum and siltstone beds (Supplementary Note 3). Moreover, catastrophic flood events with rapid clastic transportation to the

playa lake and short-duration subaerial pedogenesis are also used to interpret low χ_{lf} values of several greyish white fine sand beds (Supplementary Note 3). In contrast, a combination of increased subaerial pedogenic magnetic mineral formation, detrital magnetic mineral preservation, and limited non-magnetic dilution is used to explain high χ_{lf} values in the red mudstone beds (Supplementary Note 3). Thus, combined pedogenic, dissolution, dilution, and preservation effects are now suggested as a likely mechanism of orbital-scale χ_{lf} changes in the Lanzhou Basin fluvial-lacustrine sediments, with support from our added new magnetic measurements and morphological and mineral analyses.

Lithological variability and age uncertainties may have induced noise (non-orbital or displaced orbital signals) in the proxy record, which may have influenced the intensity and continuity of orbital signals in the spectral evolution diagrams (Fig. 4; Supplementary Figs. 14–15). However, it appears that such noise did not change the major orbital pattern that we focused on because it is observed consistently in the χ_{lf} record using the eccentricity-based astronomical time scale, the untuned magnetostratigraphy, and the original depth domain (Fig. 4; Supplementary Figs. 14–15). Furthermore, noise is reduced by our orbital tuning (Fig. 4).

We also calculated the spectral evolution of 3-point, 5-point, and 10-point running χ_{lf} means (Supplementary Fig. 17). The 3-point and 5-point running χ_{lf} means have almost identical orbital features as the original χ_{lf} record, with a clear appearance of obliquity and precession after Oi-1 (Supplementary Fig. 17). Due to larger smoothing effects, the precession signal is smoothed away by the 10-point running χ_{lf} mean, with only a weak obliquity signal left. This is now also mentioned in the main text (lines 229–230) and Supplementary Note 4 (lines 356–361).

We also tried non-linear time series analysis with the Huang-Hilbert method (see Figure 1 below). It appears that this non-linear analysis causes too much divergences from the original Lanzhou χ_{lf} data and leads to significant artificial signals. Specifically, eccentricity components (100-kyr and 405-kyr) disappeared throughout the late Eocene–early Oligocene despite their clear presence in the eccentricity-based astronomical time scale, the untuned magnetostratigraphy, and the original depth domain (Fig. 4; Supplementary Figs. 11–12). Thus, we do not include non-linear time series analysis with the Huang-Hilbert method in the revised manuscript.

Figure 1. Empirical mode decomposition components ($c_1, c_{10}, c_{20}, c_{30}, c_{40}, c_{50}, c_{60}, c_{70}, c_{80}, c_{100}$) from the original Lanzhou χ_{lf} data on the eccentricity-based astronomical time scale with the Huang-Hilbert method (Huang et al., 1998). Generally, c_1 should contain the shortest period component in the signal, while c_{10} is enough to show the longest period component in the signal (Huang et al., 1998). However, the long period 100-kyr and 405-kyr eccentricity components in the original data remain absent in c_{10} – c_{100} . This suggests that this non-linear time series analysis is not superior to the routinely used linear methods for spectral analyses of the Lanzhou χ_{lf} data.

Huang, N.E., Shen, Z., Long, S.R., Wu, M.L.C., Shih, H.H., Zheng, Q.N., Yen, N.C., Tung, C.C., Liu, H.H., 1998. The empirical mode decomposition and the Hilbert spectrum for nonlinear and non-stationary time series analysis. *Proceedings of the Royal Society A* 454, 903–995.

Comment #4

Final comment and recommendation:

My comments above should not be understood as a principle criticism against the authors' interpretations. Actually, I like the paper, but I'm perplex by the certainty and unambiguity in which it is presented. Remaining uncertainty is no reason for rejection, this would kill 99% of all papers in geoscience. My recommendation to the editor is to support the paper for publication, but to urge the authors to be more conservative and discuss/mention the uncertainties in a fair way, leaving open questions open for future work and discussion.

Response: Thank you for recommending publication of this manuscript and for providing useful, thoughtful, and constructive suggestions on age uncertainties and on the χ_f mechanism to help us further improve the paper. The age uncertainties and magnetic proxy parameters are now addressed in much more detail in the revised manuscript. We hope that you will be satisfied with our revisions.

Reviewers' Comments:

Reviewer #4:

Remarks to the Author:

In the previous version, I missed a thorough presentation of uncertainties in the determined age model, including a discussion of the consequences of the large SAR variability. In the revised manuscript, the authors present additional information on their tuning process and new rock magnetic data, the latter confirming a complex origin of the magnetic assemblage as expected for these playa deposits. The added data and discussions eventually support a sufficiently robust age framework. I therefore think the manuscript is ready for publication.

Minor points:

(1) It would be fair to state clearly that the preferred astronomical age model deviates by c. 150 kyr from the GPTS-based C13n-C12r polarity boundary. Please also reword the statement that "magnetostratigraphy has poorer age precision than the astronomical time scale" (main text l.236-237, also stated in suppl. l.312-313); it is true that the GPTS has a lower age resolution, but the precision of polarity boundaries is certainly more precise.

(2) The IRM acquisition results (suppl. Fig.7) reveal a strong dominance of hematite over magnetite (by at least two orders); thus the statement "...point to a magnetic mineralogy dominated by magnetite and hematite (suppl. l.478) is not correct. Certainly, the susceptibility record will be largely controlled by the magnetite, despite the huge hematite dominance.

(3) I very much doubt the interpretation of the X-ray mapping (suppl. Fig.11). The grain size range of the relatively larger grains can be estimated by the BSE images (smaller ones will escape due to the SEM resolution), but it is practically impossible to distinguish magnetite and hematite by EDX. If you want to identify magnetite clearly, use ferrofluid. I suggest to remove the color-coded mineral legend, it looks useless for recognizing anything.

Reply to Reviewer #4

Minor point #1

It would be fair to state clearly that the preferred astronomical age model deviates by c. 150 kyr from the GPTS-based C13n-C12r polarity boundary. Please also reword the statement that "magnetostratigraphy has poorer age precision than the astronomical time scale" (main text 1.236-237, also stated in suppl. 1.312-313); it is true that the GPTS has a lower age resolution, but the precision of polarity boundaries is certainly more precise.

Response: Thank you for this comment. We now include a statement that "The age of the C13n–C12r reversal boundary within the Oi-1 event is ~160-kyr younger than its 2012 GPTS age (Vandenberghe et al., 2012) and ~80-kyr younger than its 2020 GPTS age (Malinverno et al., 2020) (Supplementary Table 2)" in Supplementary Note 4 (lines 313–315); the brand new 2020 study on the GPTS (Malinverno et al., 2020) is accordingly included in Supplementary Information. We have removed the statement that "magnetostratigraphy has poorer age precision than the astronomical time scale" from Supplementary Information and main text.

Minor point #2

The IRM acquisition results (suppl. Fig.7) reveal a strong dominance of hematite over magnetite (by at least two orders); thus the statement "...point to a magnetic mineralogy dominated by magnetite and hematite (suppl. 1.478) is not correct. Certainly, the susceptibility record will be largely controlled by the magnetite, despite the huge hematite dominance.

Response: Thank you for pointing this out. The reviewer is correct that the IRM acquisition results (Supplementary Fig. 7) reveal a strong dominance of hematite over magnetite, so we have now changed the statement from "These rock magnetic results point to a magnetic mineralogy dominated by magnetite and hematite" to "The IRM acquisition curve analyses reveal a strong dominance of high-coercivity hematite over low-coercivity magnetite in the sediments" (lines 482–484 in Supplementary Information). We also agree with the reviewer that "the magnetic susceptibility record will be largely controlled by the magnetite, despite the huge hematite dominance". Thus, we did not state that the Lanzhou Basin magnetic susceptibility record is dominated by hematite.

Comment #3

I very much doubt the interpretation of the X-ray mapping (suppl. Fig.11). The grain size range of the relatively larger grains can be estimated by the BSE images (smaller ones will escape due to the SEM resolution), but it is practically impossible to distinguish magnetite and hematite by EDX. If you want to identify magnetite clearly, use ferrofluid. I suggest to remove the color-coded mineral legend, it looks useless for recognizing anything.

Response: Thank you for spotting this. We no longer state that this method can be used to distinguish between magnetite and hematite; hematite and magnetite have now changed to the same colour in the colour-coded mineral legend. We have also removed the grain-size distributions of magnetite and hematite estimated from the backscattered electron (BSE) images for the magnetic extracts (previous Supplementary Figure 11C) because the reviewer is correct. We also do not distinguish hematite and magnetite in the enlarged elemental X-ray BSE maps in the revised Supplementary Figure 11. X-ray dotted BSE images provide evidence for the absence of authigenic iron sulphides (greigite and/or pyrite), which is essential for our argument that no iron sulphide formation occurred during or shortly after deposition. If we removed the colour-coded mineral

legend, the X-ray dotted BSE images would be useless for making this important observation. Thus, we wish to keep it in Supplementary Figure 11.

References cited in our response above

- Malinverno, A., Quigley, K.W., Staro, A. and Dyment, J., 2020. A late Cretaceous–Eocene geomagnetic polarity timescale (MQSD20) that steadies spreading rates on multiple mid-ocean ridge flanks. *Journal of Geophysical Research*, 125: e2020JB020034, doi.org/10.1029/2020JB020034.
- Vandenbergh, N., Hilgen, F.J. and Speijer, R.P., 2012. The Paleogene Period. In: F.M. Gradstein, J.G. Ogg, M. Schmitz and G. Ogg (Editors), *The Geologic Time Scale*. Elsevier, pp. 855-922.